# Universality, criticality and complexity of information propagation in social media

Daniele Notarmuzi [1], Claudio Castellano [2,3], Alessandro Flammini[1], Dario Mazzilli [1,3] &
Filippo Radicchi [1✉]

Statistical laws of information avalanches in social media appear, at least according to existing empirical studies, not robust across systems. As a consequence, radically different processes may represent plausible driving mechanisms for information propagation. Here, we analyze almost one billion time-stamped events collected from several online platforms – including Telegram, Twitter and Weibo – over observation windows longer than ten years, and show that the propagation of information in social media is a universal and critical process. Universality arises from the observation of identical macroscopic patterns across platforms, irrespective of the details of the specific system at hand. Critical behavior is deduced from the power-law distributions, and corresponding hyperscaling relations, characterizing size and duration of avalanches of information. Statistical testing on our data indicates that a mixture of simple and complex contagion characterizes the propagation of information in social media. Data suggest that the complexity of the process is correlated with the semantic content of the information that is propagated.

[1] Center for Complex Networks and Systems Research, Luddy School of Informatics, Computing, and Engineering, Indiana University, Bloomington, IN 47408, USA. [2] Istituto dei Sistemi Complessi (ISC-CNR), Via dei Taurini 19, I-00185 Roma, Italy. [3] Centro Ricerche Enrico Fermi, Via Panisperna 89 A, Roma, Italy.
✉email: filiradi@indiana.edu

Social media have dramatically changed the way people produce, access and consume information[1], and there is increasing evidence that online discussions have the potential to impact society in unprecedented ways[2]. For example, the public debate around the COVID-19 pandemic has been accompanied by the so-called Infodemic that is affecting the outcome of the vaccination campaign by increasing hesitancy[3–5]. Also, online discussions in the Reddit channel r/wallstreetbets induced many individuals to buy GameStop shares in opposition to the shorting operation carried out by hedge funds and professional investors. As a result, the market capitalization of the company displayed an increase of more than $22 billion in just a few days[6]. It is not surprising therefore the renewed scientific interest to comprehend the mechanisms that drive information propagation.

Analyses of the propagation of information in social media reveal, at least qualitatively, similarities with other natural phenomena such as the firing of neurons[7,8] and earthquakes[9]. These processes are characterized by bursty activity patterns. The activity consists of point-like events in time, and bursts (or avalanches) of activity are defined as sequences of close-by events. Bursts are separated by long periods of low activity. Activity can be characterized at the macroscopic level by the distributions $P(S)$ and $P(T)$ of the size $S$ and the duration $T$ of avalanches[10–15]. In real-world systems $P(S)$ and $P(T)$ have a power-law decay for large value of their argument, i.e., $P(S) \sim S^{-\tau}$ and $P(T) \sim T^{-\alpha}$[7–9,12,16–18]. This property is interpreted as evidence of the system operating at, or in the vicinity of, a critical point. This statement is supported by the theory of absorbing phase transitions according to which, if the avalanche dynamics is at a critical point, then $P(S)$ and $P(T)$ must decay as power laws, see Eq. (3). Furthermore, in a process operating at criticality, the average size of avalanches with given duration must obey the hyperscaling relation $\langle S \rangle \sim T^{\gamma}$, with $\gamma = (\alpha - 1)/(\tau - 1)$[16,19,20]. The specific values of the exponents $\tau$ and $\alpha$ typically differ for classes of systems. Their actual values are fundamental for the characterization of systems into universality classes, i.e., an ontology of processes with conceptual and practical relevance[21].

Universality is the notion that nearly identical avalanche statistics are observed for a multitude of systems governed by different dynamical laws that nevertheless share some basic core mechanisms. Criticality instead refers to the fact that avalanche statistics are characterized by algebraic distributions. Classifying a system within a universality class is informative about the basic core mechanisms that drive the unfolding of the avalanches. Where information propagation (in general, and in online social media) is concerned, the issue of the existence of well-defined universality classes is far from settled. Existing analyses typically study data collected from a single source and over short observation windows. It is often found that distributions of avalanche size and duration obey power laws, but the estimated values of the exponents vary across studies: $\tau$ values range between $\tau \simeq 2$ and $\tau \simeq 4$[13,14,22–24], whereas $\alpha \simeq 3.6$[25] or $\alpha \simeq 2.5$[26,27]. Also, empirical studies reporting on correlations between size and duration of avalanches fail to find a power law[28,29]. This variability might be ascribed to multiple operative definitions of avalanches, which can be given in terms of hashtags time series[22,28] as well as reply trees or retweet chains[13,24,30]. Furthermore, regardless of the definition, the temporal resolution can affect the avalanche distribution[12,31].

As a consequence of the variability in the distributions inferred, uncertainty about representative theoretical models remains. In particular, it is an open problem to determine when and if models based on simple contagion are more appropriate to describe the spreading of information online than those based on complex contagion. Stemming from the similarity between the spreading

of disease and information, a widely accepted paradigm is that information propagates according to a simple contagion process, where only a single exposure to activity may be sufficient for its diffusion[10,13,22,28,32,33]. Simple contagion is at the core of many theoretical models of information propagation used in the literature, all displaying critical properties of the mean-field branching process (BP), i.e., $\tau = 3/2$ and $\alpha = 2$[34–37], see Methods. However, there are quite a few studies in favor of the complex contagion paradigm[38–41]. As originally introduced by Centola and Macy, in a complex contagion process the involvement of an individual in the propagation of information requires exposure from multiple acquaintances[42]. Complex contagion is exemplified by some models, such as the linear threshold model and the Random Field Ising Model[19,43] (RFIM), see Methods. Distinguishing between simple and complex contagion and, possibly, comprehending how they coexist within the same population[44], is fundamental to understand the spreading of (mis)information in online social media[38,45].

In this work, we perform a large-scale study of (hash)tags time series from Twitter, Telegram, Weibo, Parler, StackOverflow and Delicious [see Methods and Supplementary Information (SI) A for details about the data sets]. We consider a total of 206,972,692 time series. In our study, a time series consists of all posts that carry the same topic identifier, such as a hashtag on Twitter. Taken cumulatively, our time series consists of 905,377,009 events, collected over periods even longer than 10 years. The Twitter data, collected specifically for this work, are fully available together with codes to reproduce the results of this paper[46,47]. To define avalanches in a principled fashion we adopt the approach inspired by percolation theory proposed in Ref. [31], see Methods. We provide evidence that social media share universal statistics of avalanches that are well described by power-law distributions. We also develop a novel statistical technique able to determine the level of criticality and complexity of individual time series, see Methods. We find that nearly 20% of the time series are less than 5% away from criticality. These account for 53% of all events in our data sets. At the aggregate level, each social medium displays a critical behavior that is compatible with the RFIM, indicating that, plausibly, processes compatible with complex contagion may play a preponderant role in information diffusion. A more detailed analysis reveals a more nuanced scenario, where about 50% of the individual time series are better explained in terms of a complex rather than a simple contagion process. A qualitative analysis of the most popular hashtags suggests that information concerning conversational topics, e.g., music or TV shows, spreads according to the rules of simple contagion, whereas information concerning political/societal controversies shows signatures of an underlying complex contagion process.

## Results

**Selection of temporal resolution.** Here, an avalanche is defined as a maximal subset of contiguous events in a time series such that two consecutive ones are separated by a time interval smaller than $\Delta$. A proper choice $\Delta^*$ of the time resolution $\Delta$ for the specific data set at hand is necessary to avoid significant distortion in the resulting avalanche statistics. This is true for synthetic time series generated by temporal point processes[31], but also for the empirical time series as those analyzed in this paper (see SI E for details). To determine the value of $\Delta^*$ we use the principled method developed in Ref. [31] that identifies $\Delta^*$ as the critical point of a one-dimensional percolation model, see Methods for details. Results are presented in Fig. 1. Values of $\Delta^*$ for each data set are reported in the SI A; they vary substantially across data sets, from $\Delta^* \simeq 1500$ s for Twitter to $\Delta^* \simeq 30,000$ s for Telegram (Fig. 1b).

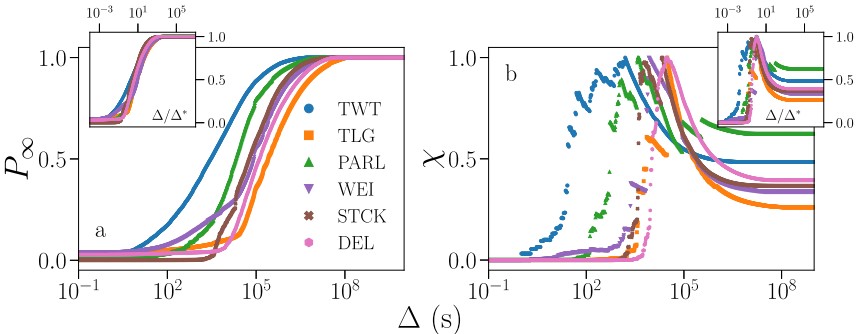

**Fig. 1 Universality of information propagation in online social media. a** In the main panel, we show the percolation strength $P_\infty$ as a function of the temporal resolution $\Delta$, see Eq. (1). The unit of measurement for $\Delta$ is one second. Different colors/symbols refer to different social media: Twitter (TWT), Telegram (TLG), Parler (PARL), Weibo (WEI), StackOverflow (STCK), and Delicious (DEL). In the inset, we plot the same data as in the main panel, but with the horizontal axis rescaled as $\Delta \rightarrow \Delta/\Delta^*$. **b** In the main panel, we plot the susceptibility $\chi$ as a function of the time resolution for the same data as in A, see Eq. (1). The optimal resolution $\Delta^*$ is identified as the location of the peak of the susceptibility, see Eq. (2). In the inset, we plot the same data as in the main panel, but with the rescaling $\Delta \rightarrow \Delta/\Delta^*$. For the sake of comparison, each curve has been normalized to its maximum $\chi^*$, see Methods.

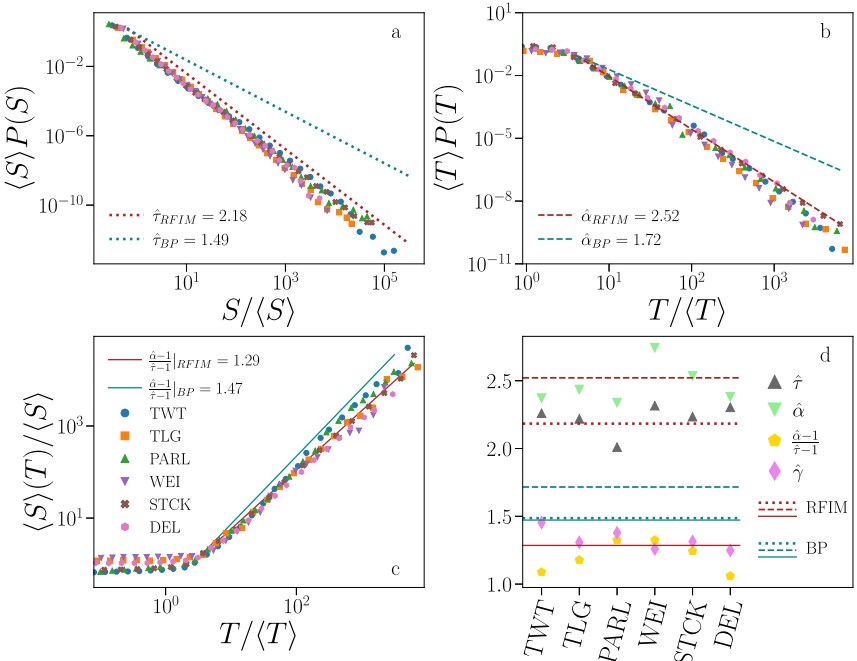

**Fig. 2 Universality and criticality of information propagation in social media. a** Avalanche size distribution. Different colors/symbols indicate data obtained from different social media. Acronyms are defined as in Fig. 1. Dotted lines represent the maximum likelihood estimators of the exponent $\tau$ obtained by fitting the Random Field Ising Model (RFIM), in red, and the branching process (BP), in teal. The RFIM was fitted using $N = 10^9$, $R = 0.8$ and considering the same number of avalanches as the Twitter sample. The BP was fitted using $n = 1.0$ and sampling $10^6$ avalanches. Distributions are displayed via logarithmic binning of the data. To make distributions collapse one on the top of the other, the size is multiplied by the factor $1/\langle S \rangle$ and probabilities are multiplied by the factor $\langle S \rangle$. Cumulative distributions are reported in SI E. **b** Distribution of avalanche duration for the same data as in panel **a**. To make distributions collapse one on the top of the other, duration is multiplied by the factor $1/\langle T \rangle$ and probabilities are multiplied by the factor $\langle T \rangle$. Dashed lines represent the maximum likelihood estimators of the exponent $\alpha$ obtained by fitting the RFIM (red) and the BP (teal). **c** Average size of avalanches with given duration. Data are the same as in **a** and **b**. To make the curves collapse one on top of the other, the abscissa of each curve is rescaled as $T/\langle T \rangle$ and the ordinate is rescaled as $\langle S \rangle (T)/\langle S \rangle$. Solid lines represent the hyperscaling exponent $(\hat{\alpha} - 1)/(\hat{\tau} - 1)$ obtained using the maximum likelihood estimators of $\tau$ and $\alpha$ for the RFIM (red) and for the BP (teal). **d** Maximum likelihood estimates of the exponents $\hat{\tau}$, $\hat{\alpha}$ and $\hat{\gamma}$, see SI C for details. We also display the ratio $(\hat{\alpha} - 1)/(\hat{\tau} - 1)$. Error bars are always smaller than the size of the symbol. Dotted lines correspond to the best fit of the exponent $\tau$ to the RFIM (red) and to the BP (teal), as shown in panel **a**. Analogously for dashed lines, representing the best fit of $\alpha$ shown in **b** and for solid lines, representing the hyperscaling relation shown in **c**.

Once the time resolution is rescaled according to $\Delta \rightarrow \Delta/\Delta^*$, the curves of the percolation strength for the different data sets exhibit a nearly identical quantitative behavior, see insets of Fig. 1. This fact suggests the possibility of seeing the propagation of information in social media as a universal process, with $\Delta^*$ representing the natural resolution for observing information avalanches. Figure 2a, b shows the distributions of avalanche size and duration obtained by setting $\Delta = \Delta^*$. Figure 2c shows the relation between average size and duration. The collapse of the curves relative to different data sets on a single curve hints once more, at least when data are considered at the aggregate level, to processes belonging to the same universality class.

**Criticality and universality of avalanche statistics**. The avalanche statistics of Fig. 2a–c seems well described by power laws, indicating that the underlying process is (nearly) critical, and that its universality class can be identified by estimating the value of the critical exponents $\tau$, $\alpha$, and $\gamma$, see Eq. (3)[21]. We rely on maximum likelihood estimation for $\tau$ and $\alpha$[48]; linear regression on the logarithm of the relation $\langle S \rangle \sim T^\gamma$ is used to estimate $\gamma$. Results are reported in Fig. 2d, see SI C for details. The estimated exponent $\hat{\tau}$ is compatible with the one of the mean-field RFIM universality class, i.e., $\tau = 9/4$[19]. The compatibility of the avalanche statistics with those of a homogeneous mean-field model is not surprising given that in some social media there is no underlying network among users and in others there are mechanisms for the propagation of information that bypass it. For example, in Telegram all users who subscribe to a channel receive all messages sent from any other user of that channel, meaning that there is an all-to-all network among all users of the channel as in the mean-field version of the RFIM. In Stack-Overflow there is no underlying network as users do not follow each other, rather they search for content using common tools offered by the platform. Even in Twitter, where users have follower–followee relationships, the network can be easily bypassed by the way the platform manages users' feeds. There is an apparent mismatch between our estimates $\hat{\alpha}$ and $\hat{\gamma}$ and the RFIM predictions $\alpha = 7/2$ and $\gamma = 2$ due to finite-size effects. To properly address this issue, we performed numerical simulations of the RFIM, and measured the maximum likelihood estimators of $\tau$ and $\alpha$. For consistency, we performed the same operation for the BP too. The results of Fig. 2 reveal that, overall, our data are compatible with the phenomenology of the RFIM and not with the phenomenology of the BP.

The proximity of exponents estimated across different data sets points to the existence of a genuine and distinctive universality class for information propagation in social media when considered at the aggregate level. In particular, this class seems to be different from that of the BP often invoked as a representative in phenomena related to information diffusion. This universal scaling is a genuine feature of social media, as if we repeat the same analysis on time series describing activity in very different types of systems, e.g., brain networks and earthquakes, avalanche duration and size still decay in a power-law fashion, but with radically different exponent values, see SI D for details. In particular, for neuronal avalanches in the brain, we recover exponents compatible with previous studies[8,49–51].

**Complexity of avalanche statistics**. To assess if the statistical properties obtained on aggregate data are representative of individual time series, we develop a maximum likelihood method to fit the time series against the BP and the RFIM. The technique is inspired by the work of Ref. [48], see Methods for details. The method supports three different tests. First, it establishes the regime of a time series, depending on how the best estimate of the branching ratio parameter $\hat{n}$ compares to the critical value $n_c = 1$ for the BP, or how the best estimate of the disorder parameter $\hat{R}$ compares to the critical value $R_c = \sqrt{2/\pi} \simeq 0.8$ for the RFIM. Second, it evaluates the goodness of the individual fits via their $p$ values. Similarly to the prescription of Ref. [48], we set the threshold for statistical significance equal to $p = 0.1$. We verified, however, that the outcome of the analysis is not greatly affected by the choice of the threshold value, see SI J. Third, it establishes whether a time series is better modeled by the BP or by the RFIM by comparing their likelihood.

Results of our analysis are reported in Figs. 3 and 4. Our method is applied only to time series that contain at least two avalanches larger than $S_{min} = 10$. These two avalanches must also

have different sizes, so that $P(S)$ has at least two non-zero values. Tests of robustness for different $S_{min}$ values are reported in the SI J. In all systems we find that the best fitting parameter assumes values over a broad range, encompassing a large portion of the subcritical phase and the critical point of the models (Fig. 3a, b). The majority of events belongs to a minority of time series giving rise to the largest avalanches. As a consequence, the large-scale behavior of each system is mainly determined by those few time series that are fitted in a narrow region of the parameter space close to the critical point for both the BP and the RFIM (insets of Fig. 3a, b). Also, our tests indicate that the vast majority of time series are well described by at least one of the two models (Fig. 4a). The model selection indicates that individual time series are divided into two nearly equally populated classes, one better described by the BP and the other by the RFIM (Fig. 4a). Simple and complex contagion thus coexist in social media, with only a mild dominance of complex over simple contagion (Fig. 3c). The individual-level analysis is not incompatible with the results obtained for the aggregate data (Fig. 2). If we aggregate data only from the time series that we attributed to the class of complex contagion, we consistently recover a power-law scaling compatible with that class for all avalanche sizes, see Fig. 3d. However, the aggregation of time series that are classified in the BP class generate a distribution characterized by a neat crossover from BP scaling for small avalanches to RFIM scaling for large avalanches (Fig. 3d). The mixture produces a universal distribution that is overall more compatible with the RFIM universality class rather than the BP class (Fig. 2c).

## Discussion

We showed that temporal patterns characterizing bursts of activity in online social media are conveniently classified in two universality classes. This finding suggests that few core mechanisms determine the large-scale behavior of information diffusion and that many peculiarities that characterize individual platforms are far less relevant. Also, in contrast with the vast majority of previous studies where purely diffusive models have been considered[37], we showed that information propagation in social media is often better described by complex contagion dynamics. Complex contagion is here exemplified by the RFIM, an agent-based model of activation originally formulated to describe the para-to-ferromagnetic phase transition in metals[19]. Recast in the language proper to the description of information propagation[52], the RFIM prescribes that each agent (i) has a personal opinion, (ii) is subject to the social influence exerted by the agents she interacts with, and (iii) is also driven by an external force representing the public information about exogenous events. These appear reasonable assumptions for modeling many realistic discussions happening in social media. Figure 4b shows the 30 most popular Twitter hashtags identified by our method either in the simple or in the complex contagion classes. In the category of simple contagion, we find conversational topics, mostly related to music or cinema/TV shows. Hashtags belonging to the class of complex contagion display either periodic patterns or are related to political/controversial themes. This suggests the existence of a relation between the semantics of hashtags and the universality class of the corresponding time series. This qualitative picture fits with previous studies that have explicitly focused on the semantic of different hashtags in Twitter[45]. For both classes of information avalanches, we inferred the dynamics underlying their generation as critical, a fact that provides theoretical ground for the surprising but remarkable robustness of our findings. The presence of a large portion of social media content that acquires popularity via complex contagion dynamics calls for a reconsideration of predictive algorithms relying on the temporal characteristics of

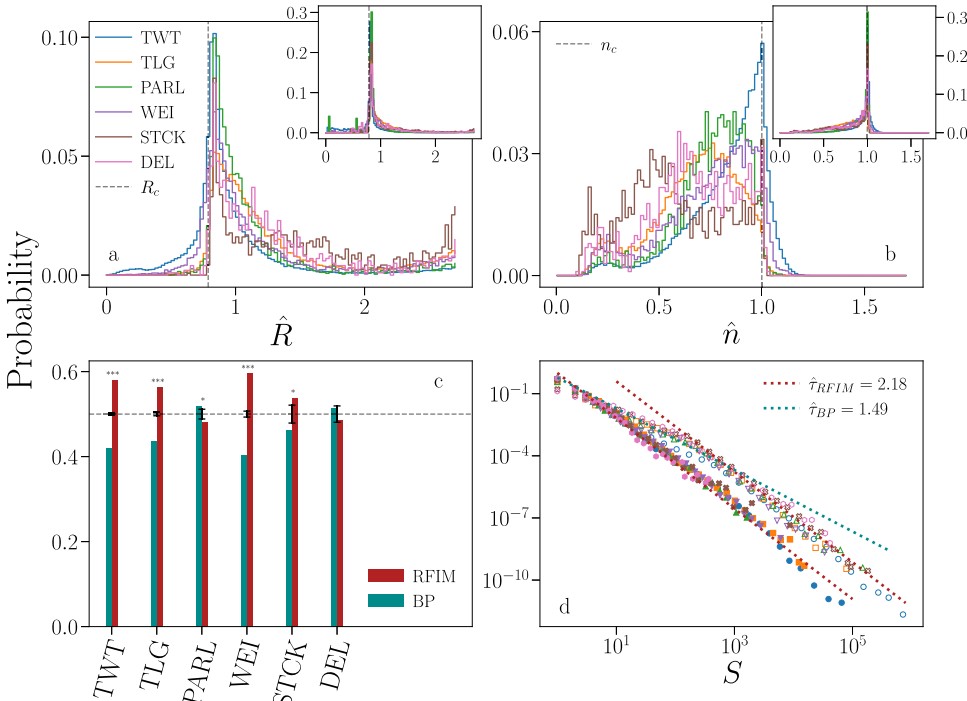

**Fig. 3 Criticality and complexity of information propagation in online social media. a** We fit each individual time series against the Random Field Ising Model (RFIM) to determine the best estimator of the disorder parameter $\hat{R}$. We then compute the distribution of $\hat{R}$ for all time series of a given data set. Acronyms of the data sets are defined as in Fig. 1. We fit only avalanches whose size is at least equal to $S_{min} = 10$. The dashed vertical gray line denotes $R_c$, i.e., the critical value of the RFIM parameter. The inset shows the same data as in main panel, but each time series contributes to the histogram with a weight equal to its total number of events. **b** Same analysis as in **a**, but obtained by fitting individual time series against the branching process (BP) to determine the best estimator of the branching ratio $\hat{n}$. **c** Probability that the log-likelihood ratio test favors the RFIM over the BP (teal), or vice versa the BP over the RFIM (red). Only time series that are sufficiently well fitted by both models are considered in the analysis, see Fig. 4b. Error bars represent $\sigma/N$, where $N$ is the sample size and $\sigma = \sqrt{0.25\,N}$ is the standard deviation of a binomial distribution with probability of success equal to 1/2. Asterisks are used to denote significant deviations from the unbiased binomial model, i.e., three asterisks indicate $p < 0.001$, and one asterisk stands for $p < 0.1$. **d** We use the classification of panel **c** to divide time series in two distinct classes. We then consider only time series whose best estimators are sufficiently close to the critical value of the model representing their class, i.e., $|\hat{R} - R_c|/R_c \leq 0.05$ or $|\hat{n} - n_c|/n_c \leq 0.05$, to compute the distribution of avalanche size for each class. Full symbols are used for the RFIM class, empty markers are used to display the distributions of the BP class. The dotted lines correspond to the best fit of the exponent $\tau$ to the RFIM (red) and to the BP (teal).

the signal only, because these algorithms often neglect the semantics of hashtags and, even more frequently, the characteristics of the network over which they spread[53–57]. Both aspects are important for the successful characterization of the process underlying the propagation of information[38,45,58,59]. We further speculate that our results extend beyond the six platforms considered here. If so, there must be a mechanism that explains the universality shown by the data, involving critical dynamics that is independent of the peculiarities implemented in the individual platforms. Understanding where this mechanism is rooted in and how to exploit this mechanism for the prediction of the propagation of information in online social media remain open challenges for future research.

## Methods
**Data**. We build a time series for each (hash)tag appearing in the data at our disposal. A time series contains the times, i.e., $\{t_1, t_2, \ldots\}$, when the (hash)tag is observed in the data.

Specifically, the Twitter data set is composed of 2,353,192,777 tweets corresponding to a 10% random sample of all Tweets posted on Twitter during the observation window from October 1 to November 30, 2019. The collection of this data has been performed via the Indiana University OSoME Decahose stream[60,61]. Telegram time series are extracted from a total of 317,224,715 messages, originally collected in Ref. [62]. Parler time series are extracted from a total of 183,062,974 posts, originally collected in Ref. [63]. Weibo time series are extracted from 226,841,249 posts, originally collected in Ref. [64]. StackOverflow time series are extracted from a total number of 46,947,635 questions and answers. Delicious time series were extracted from 7,034,524 users actions, originally collected in Ref. [65].

Timestamps always have the temporal resolution of the second, except for the StackOverflow data set, whose temporal resolution is the millisecond.

We pre-process the data so that the number of events per unit time is roughly constant over the whole temporal window considered (see SI A for details) to obtain a corpus of 206,972,692 time series consisting of 905,377,009 total events.

**Selection of the temporal resolution**. We follow the same procedure as in Ref. [31]. Given a time series $\{t_1, t_2, \ldots\}$, we define an avalanche starting at $t_b$ as a sequence of events $\{t_b, t_{b+1}, \ldots, t_{b+S-1}\}$ such that $t_b - t_{b-1} > \Delta$, $t_{b+S} - t_{b+S-1} > \Delta$ and $t_{b+i} - t_{b+i-1} \leq \Delta$ for all $i = 1, \ldots, S$, where $\Delta$ is the resolution parameter. The size $S$ of an avalanche is the number of events within it and the duration $T$ is the time lag between the first and last event in the avalanche, i.e., $T = t_{b+S-1} - t_b$. Depending on the value of $\Delta$, the same time series is composed of different avalanches.

We identify the optimal resolution $\Delta^*$ as the critical point of a one-dimensional percolation model that is used to describe the time series. Each time series in a data set is considered as an instance of the one-dimensional percolation model. We measure the size $S_M$ of the largest avalanche within each time series. We define the percolation strength $P_\infty$ and its associated susceptibility $\chi$, respectively, as

$$P_\infty = \langle S_M \rangle$$
$$\chi = \frac{\langle S_M^2 \rangle - \langle S_M \rangle^2}{\langle S_M \rangle}, \tag{1}$$

where $\langle S_M \rangle$ and $\langle S_M^2 \rangle$ are, respectively, the first and second moments of the distribution of the size of the largest avalanche $S_M$ across all time series in a data set. $\Delta^*$ is computed as the resolution maximizing $\chi$, i.e.,

$$\Delta^* = \arg\max \chi(\Delta) . \tag{2}$$

As time series with only one event introduces an offset in the measure of $P_\infty$ and are not informative with respect to the optimal resolution $\Delta^*$, i.e., $S_M = 1$ for any $\Delta$ in these time series, we remove them from the sample and compute $P_\infty$ and $\chi$ considering only time series composed of at least two events.

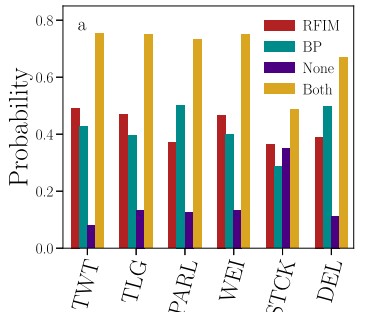

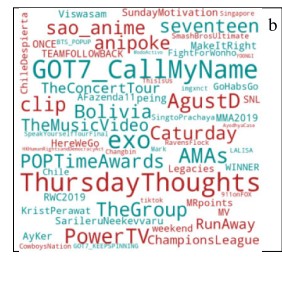

**Fig. 4 Simple vs. complex contagion in online social media. a** We consider avalanches with size $S \geq S_{min} = 10$ and fit them against the branching process (BP) and the Random Field Ising Model (RFIM). For each time series, we establish whether the fits against the individual models are statistically significant or not; if both fits cannot be rejected, we then select the best model by means of the log-likelihood ratio. We report the fraction of time series that are classified in the RFIM class. This fact may happen because the RFIM fit cannot be rejected whereas the BP is rejected, or both fits cannot be rejected but the RFIM is favored over the BP in terms of log-likelihood ratio. The fraction of time series that are classified as BP is defined in an analogous manner. The fraction of time series that is classified as neither BP nor RFIM is represented by the bar labeled as "None." Finally, some time series pass both statistical tests. Their fraction is denoted by the label "Both" in the figure. In this case, the log-likelihood ratio test is required for model selection, see Fig. 3c. **b** We restrict our attention to Twitter hashtags containing characters from the English alphabet only, and display the 30 most popular hashtags classified either in the RFIM (blue) or the BP (red) classes. The font size is proportional to the rank of the hashtag in each class. Hashtags of both classes are selected among those that are sufficiently critical, i.e., $|\hat{R} - R_c|/R_c \leq 0.05$ for a time series in the RFIM class or $|\hat{n} - n_c|/n_c \leq 0.05$ for a time series in the BP class.

Values of the optimal resolution $\Delta^*$ are reported in SI A. Note that the avalanche statistics reported in Fig. 2 is obtained considering all avalanches, excluding the largest one of each time series. This choice is due to the well-known fact that in percolation theory the largest cluster respects different statistics than that of finite clusters[66].

**The branching process**. In the BP an individual initially active spreads activity to a random number of peers, who can in turn spread activity further[34]. The process continues for a number $T$ of time steps or generations, until there is a generation in which no individual further spreads activity. $T$ is the duration of the avalanche. The size $S$ of the avalanche is the total number of individuals activated during the avalanche. The average number of individuals who are activated from a single spreader is the branching ratio $n$ and the model is critical for $n = n_c = 1$. The branching ratio is the only tunable parameter of the model.

Finite avalanches of activity in the BP obey the laws

$$P(S) = S^{-\tau} \mathcal{D}_S(S^\sigma n')$$
$$P(T) = T^{-\alpha} \mathcal{D}_T(T^{1/z\nu} n') \quad (3)$$
$$\langle S \rangle(T) \propto T^\gamma ,$$

where $\langle \cdot \rangle$ is the average over different avalanches, and $P(S)$ and $P(T)$ are the probability distributions of $S$ and $T$, respectively. The functions $\mathcal{D}_S$ and $\mathcal{D}_T$ are known as scaling functions and introduce corrections at small values of their argument, where we have defined the reduced distance from the critical point $n' = |n - n_c|/n_c$. The BP is characterized by the exponent $\tau = 3/2$, $\alpha = 2$ and $\gamma = 2$. The above exponents are not independent, rather they are related by $\gamma = 1/(\sigma z \nu) = (\alpha - 1)/(\tau - 1)$. $\sigma$, $z$ and $\nu$ are additional critical exponents that we do not explicitly consider in our analysis.

**The Random Field Ising Model**. We consider the mean-field formulation of the zero-temperature RFIM. Agent $i$ is characterized by the state variable $y_i = \pm 1$ indicating whether the agent is active, $y_i = +1$, or not, $y_i = -1$. Each agent $i$ has a propensity $h_i$ to become active, with $h_i \in (-\infty, +\infty)$. A large value of $h_i$ indicates that the agent is particularly prone to become active. Agents interact by means of ferromagnetic interactions that model social pressure, i.e., active neighbors push an inactive agent to become active. The whole system is further affected by public information that all agents have access to and that pushes users toward becoming active with intensity $H \in (-\infty, +\infty)$. In the initial configuration, all agents are

inactive. The external pressure $H$ grows till the agent with the largest $h_i$ value becomes active. This change of state can trigger an avalanche of activity in the other nodes. Specifically, agent $j$ becomes active if the following condition is met

$$H + h_j + N^{-1} \sum_{k \neq j} y_k > 0 , \quad (4)$$

where $N$ is the system size and the mean-field formulation is expressed by the all-to-all interaction. Once in the active state, agents cannot change their state back to inactive. When an avalanche ends, the external pressure $H$ grows again until a new user becomes active and triggers a new avalanche. The field is frozen during the unfolding of avalanches, meaning that avalanches are characterized by a time scale much shorter than the one characterizing external pressure. In the long-term limit, when $H = +\infty$, all agents become active. The size $S$ of an avalanche is given by the number of users that are activated during the avalanche; its duration $T$ is given by the activation rounds characterizing the avalanche.

The stochasticity of the model comes from the random nature of the propensities $h_i$, extracted from a normal distribution with zero mean and variance $R$. The choice of the normal distribution is quite standard both for ferromagnets and social systems[52]. $R$ is the only tunable parameter of the model, and the model is critical for $R = R_c = \sqrt{2/\pi}$. Avalanche statistics obey laws similar to those of Eq. (3). The functional form of the scaling functions, however, is not the same as in the BP; also, their argument is given in terms of the distance from the critical point of RFIM, i.e., $n' = |n - n_c|/n_c$ is replaced by $R' = |R - R_c|/R_c$. The values of the critical exponents are $\tau = 9/4$, $\alpha = 7/2$ and $\gamma = 2$[19]. In SI F, we show that the peculiar form of the scaling function $\mathcal{D}_T$ introduces strong preasymptotic corrections to the functions $P(T)$ and $\langle S \rangle (T)$, affecting the measure of $\alpha$ and $\gamma$ obtained through numerical simulations of the model.

**Model selection**. To ascribe each time series to a dynamical model, we first fit each model individually by maximizing its likelihood. We evaluate the $p$ value of the fits and, if both hypotheses cannot be rejected, we select the best fit via the log-likelihood ratio test.

To perform the fit, we compare the probability distribution $P(S)$ of the avalanche sizes identified in the time series with the conditional distributions of the avalanche size $Q_{RFIM}(S|R)$ and $Q_{BP}(S|n)$, respectively, obtained for the RFIM and the BP for a given value of the parameters $R$ and $n$. The construction of the model distributions $Q$ requires discretizing the parameter space of the models. In this study $R$ varies in the interval [0.025, 2.7] by steps of length $dR = 0.025$ and $n$ varies in [0.02, 1.7] by steps of length $dn = 0.015$. $dR$ ($dn$) represents the uncertainty on the parameter. Instead of sampling avalanches from the model at a precisely given value of $R$ ($n$), we consider model instances corresponding to $R$ ($n$) values uniformly distributed over an interval of length $dR$ ($dn$) centered at $R$ ($n$). The distribution $Q$ corresponding to a specific value of the parameter model is constructed as the superposition of 500 distributions whose parameter values are randomly sampled from the corresponding interval. Fitting a time series to a model means estimating the best parameter with an accuracy of $dR$ ($dn$) for the RFIM (BP).

Given the empirical distribution $P$ and the model distributions $Q$, we evaluate the log-likelihood function

$$L(P||Q) = \sum_{S \geq S_{min}} P(S) \log[Q(S)] . \quad (5)$$

The summation is performed over all avalanches with $S \geq S_{min}$, a parameter we vary in our analysis. The distributions $P$ and $Q$ are normalized over the interval $[S_{min}, \infty)$ to account for this fact. The best fit is obtained by finding the parameter value that maximizes the log-likelihood of Eq. (5). The maximization of the log-likelihood of Eq. (5) is equivalent to the minimization of the cross-entropy of the distribution $Q$ relative to the distribution $P$. To avoid numerical problems in the estimation of the likelihood, we smoothen the function $Q$. Details are provided in SI G.

To assign a $p$ value to a fit, we follow the prescription of Ref. [48]. Indicating with $Z_{tail}/Z$ the fraction of avalanches with $S \geq S_{min}$ in the fitted time series, a synthetic sample of $Z$ avalanches is created by sampling avalanches with $S \geq S_{min}$ from the selected model $Q$ with probability $Z_{tail}/Z$ and by sampling avalanches with $S < S_{min}$ from the empirical distribution with complementary probability. Each of these synthetic samples is fitted analogously to the original sample obtained from the time series. We compute the Kolmogorov–Smirnov (KS) distance between the empirical distribution $P$ and the selected model $Q$, as well as between the synthetic samples and their best fit. The $p$ value of the fit is defined as the fraction of synthetic samples whose KS distance from the selected model is larger than the KS distance between the real sample and its best model. The hypothesis that the sample has been generated by a certain dynamical model, say the RFIM, cannot be rejected if the $p$ value of the fit to the RFIM is larger than a pre-established significance threshold. We set the threshold to 0.1 in the main text, following the prescription of Ref. [48]. Tests of robustness against the choice of this parameter value are reported in SI J.

If one of the two hypotheses can be rejected but the other cannot, the non-rejected model automatically becomes the selected one. If both hypotheses can be rejected, the time series is classified as "None." If, however, both hypotheses cannot be rejected, we select as the best model the one with the largest likelihood[48]. We neglect the possibility that a single time series could be described by a mixture of

models. Empirical data are fitted only if the time series contains at least 50 events and at least 10 avalanches.

We validate our fitting procedure applying it to synthetic distributions $P$ generated by the RFIM or by the BP. Results are shown in SI I and confirm the ability of our procedure to identify the ground-truth model and the correct value of the parameter.

More details about the fitting and model selection protocol, including tests of robustness against the threshold on the $p$ value and on $S_{min}$, are given in the SI.

**Reporting summary**. Further information on research design is available in the Nature Research Reporting Summary linked to this article.

## Data availability

The Twitter data generated in this study have been deposited in the Zenodo (https://zenodo.org/record/5779063#.Yg_aP-7MLCV) and GitHub (https://github.com/DaniMuzi/SocialMedia) database[46,47]. Telegram, Parler, Weibo, StackOverflow, and Delicious data used in this study have been generated in other works. URLs to each of these data sets are provided in SI A.

## Code availability

The Python and C codes used for this project are available on Zenodo (https://zenodo.org/record/5779063#.Yg_aP-7MLCV) and GitHub (https://github.com/DaniMuzi/SocialMedia)[46,47].

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

## Acknowledgements

F.R. acknowledges support from the National Science Foundation (CMMI-1552487). D.N. was partially funded by the National Science Foundation NRT grant 1735095. Any opinions, findings, and conclusions or recommendations expressed in this work are those of the author(s) and do not necessarily reflect the views of the National Science Foundation. A.F. acknowledges support from DARPA award HR001121C0169.

## Author contributions

D.N., C.C., A.F., D.M., and F.R. designed the experiments and wrote the paper. D.N. performed the data collection and the experiments.

## Competing interests

The authors declare no competing interests.
