## [Peer Review File · Nature Communications]

REVIEWER COMMENTS

Review of NCOMMS-21-37625

Last compiled on 2021-10-28

This manuscript analyzes (hash)tags time series extracted from six online social media platforms and shows how cascades, when properly temporally circumscribed (parameter Δ^*), display a universal scaling in the distribution of their size (S) and duration (T), as well as in the relationship between their expected size ($\langle S \rangle$) given their duration.

I find the analysis sound, and the results are compelling and interesting. I particularly like how the results support a distinction between cascades on Twitter based on semantic, hinting that some themes cascade similarly to simple contagion, while other more controversial themes cascade with properties akin to complex contagion. I agree with the manuscript that such observation could have an important impact on our collective fight against the spread of misinformation on social media.

While the analysis and the novelty of the results are interesting enough to be granted publication in *Nature Communications* eventually, I believe that some work is still required to bring the manuscript up to the proper standards (see comments below). For instance, I do not find the manuscript to be sufficiently self-contained in the sense that one needs to refer to the Supplementary Material on many occasions to understand the results presented in the main text. More details about the datasets, models and analysis should therefore be moved to the main text (or to the *Material and Methods* section).

I will be willing to reconsider the manuscript, once my comments listed below have been addressed.

Major comments

(critical brain dynamics): The manuscript says in its introduction that: *“there is large agreement on the fact that neuronal activity in the brain is universal and critical”*. After consulting with colleagues working in neuroscience, it seems that the concept of criticality in the brain is indeed quite popular among scientists with a physics-related background, but that it is still debated among scientists from the other realms of neuroscience. For instance, Ref. [9] says

This hypothesis is suggestive as it proposes that criticality could constitute a general and common organizing strategy in biology stemming from the physics of phase transitions. However, despite its implications, this is still in its infancy state as a well-founded theory and, as such, it has elicited some skepticism.

As another example, Ref. [5] says

Thus this hypothesis, and in particular the experimental work by Beggs and Plenz (2003), has triggered an avalanche of research, with thousands of studies referring to it. Nonetheless, experimental results are still contradictory.

While I believe that recent results like those of Ref. [4] are quite interesting, I nevertheless think the manuscript would be best served by depicting a more nuanced picture of the current state of the debate.

(neuronal activity vs simple contagion): The manuscript should expand on how the observation that the critical exponent values measured for neuronal activity *“are those of the universality class of the mean-field branching process”* and therefore that neurons *“influence each other according to a simple contagion process”* compare with *temporal and spatial summation mechanisms* which are reminiscent of a threshold model and thus of

complex contagion. Indeed, it is my understanding that the universality observed in scaling analysis usually comes at the price of not ignoring the “microscopic” details of the dynamics [7]. Hence, it is one thing to say that neuronal activity belongs to the same universality class as BP, but it is another to say that “[n]euronal activity may be modeled as a simple contagion process [...]”.

(Datasets, BP and RFIM): Elements from sections A, B, D, E and F should find their way in the *Material and Methods* section of the manuscript.

(Deviation from RFIM): On page 3, the manuscript says

At the aggregate level, each social media displays a critical behavior that is compatible with the RFIM, indicating that, plausibly, information propagates in social media according to a complex contagion process.

but later says

There is an apparent mismatch between our estimates $\hat{\alpha}$ and $\hat{\gamma}$ and the RFIM predictions $\alpha = 7/2$ and $\gamma = 2$. The mismatch can be theoretically explained by the peculiar shape of the scaling function characterizing the distribution of avalanche duration, which affects also the estimate of $\hat{\gamma}$ [52]. Difficulties in observing the asymptotic exponents of the RFIM due to the effect of the scaling functions emerge also in numerical simulations of the RFIM and are well known [19].

Looking at these two references [3, 7], it was not at all clear to me why $\hat{\alpha} = 5/2$ shown on Fig. 1 should be considered as coherent with RFIM for which $\alpha = 7/2$ (and consequently for $\hat{\gamma} = 6/5$ and $\gamma = 2$). This justification should be better explained and illustrated.

(Figure 1): The meaning of “*percolation strength*” is not straightforward in this context. I understand that the manuscript adopts the approach presented in Ref. [6] (from the same authors) but for the sake of completeness, and because the approach is not standard yet, the clarity of the manuscript would greatly improve if a description of the method (including the role of the susceptibility) were briefly explained in a *Material and Methods* section. Section D of the SM could also be moved to the manuscript, but I believe a shorter note would be sufficient.

(Figure 1): The scalings obtained for BP and RFIM should appear on every subplot (A, B and C), and the scaling found using the method from Ref. [2] should be shown using another line style and color.

(natural resolution): The manuscript states that “[...] Δ^* representing the natural resolution for observing information avalanches.” Do the authors have thought about possible explanations for the substantial variation of these time scales from one dataset to the other (i.e. “from $\Delta^* \simeq 1500$ s for Twitter to $\Delta^* \simeq 30000$ s for Telegram”).

(compatibility with mean-field model): It would be instructive if the manuscript could expand more on the claim that

The compatibility of avalanche statistics with those of a homogeneous mean-field model is not surprising given that in some social media there is no underlying network among users and in the others there are mechanisms for the propagation of information that bypass it.

by providing concrete examples. For instance, what mechanism allows the cascade of a hashtag to bypass the underlying network structure of Telegram?

(homogeneous mean-field model): It is not clear from the manuscript alone why the RFIM is a “*homogeneous mean-field model*”.

(fitting power laws): It would be instructive if the authors explained why they used the methods of Ref. [2] instead of the more recent approaches proposed in Refs. [1, 8].

(Page 3): Why did the authors considered “*avalanches that contain at least two avalanches larger than S_{\min}* ”. Is the method robust if they chose other values (including 1) instead of 2?

(Page 4): The manuscript says “[...] *is subject to the social influence exerted by the agents she interacts with [...]*” when, in the RFIM, the agent interacts with *everyone*. Does this difference, from a network to a mean-field approach, make a difference?

(Equation S4): Equation (S4) should read

$$\hat{\eta} = 1 + Z \left(\sum_{i=1}^Z \ln \frac{x_i}{x_{\min}} \right)^{-1}$$

Minor comments

(Introduction): I would suggest incorporating the footnote into the main text. Otherwise the first paragraph feels somewhat superficial.

(Page 4): Something is missing in “*are so basic that underlie information diffusion*”.

(Page 4): What “*topological features*” are the authors referring to here?

(Code on Gihub): Have the authors considered archiving the repository containing the Twitter data using services like Zenodo to ensure its long-term availability?

(Figures 1, 2 and 3): Please add the symbol associated with the quantity whenever possible. For instance, “*Size*”, “*Rescaled duration*” and “*Disorder*” should read “*Size (S)*”, “*Rescaled duration (T/Δ^*)*” and “*Disorder (R)*”.

(Figure 2): Would it be useful to display the raw distributions $P(S)$ and $P(T)$ in an inset?

(Figure 3C): The color used for BP [RGB: (0, 139, 139)] is closer to teal [RGB: (0, 128, 128)] than to blue.

(Figure 4): A table or an histogram would be a better choice than a word cloud where the relative rank between word is biased by their length.

(Bibliography): Adding the title of the references (and a clickable link) would greatly help the reader to follow the literature the manuscript builds upon. If using the `revtex` document class, simply add the option `longbibliography` and include the `doi` field in the `.bib` file. The package `hyperref` may also be required.

References

- [1] A. D. Broido and A. Clauset, *Scale-free networks are rare*, Nat. Commun. **10**, 1017 (2019).
- [2] A. Clauset, C. R. Shalizi, and M. E. J. Newman, *Power-Law Distributions in Empirical Data*, SIAM Rev. **51**, 661–703 (2009).
- [3] S. di Santo, P. Villegas, R. Burioni, and M. A. Muñoz, *Simple unified view of branching process statistics: Random walks in balanced logarithmic potentials*, Phys. Rev. E **95**, 032115 (2017).

- [4] L. J. Fosque, R. V. Williams-García, J. M. Beggs, and G. Ortiz, *Evidence for Quasicritical Brain Dynamics*, *Phys. Rev. Lett.* **126**, 098101 (2021).
- [5] M. A. Muñoz, *Colloquium: Criticality and dynamical scaling in living systems*, *Rev. Mod. Phys.* **90**, 031001 (2018).
- [6] D. Notarmuzi, C. Castellano, A. Flammini, D. Mazzilli, and F. Radicchi, *Percolation theory of self-exciting temporal processes*, *Phys. Rev. E* **103**, L020302 (2021).
- [7] J. P. Sethna, K. A. Dahmen, and C. R. Myers, *Crackling noise*, *Nature* **410**, 242–250 (2001).
- [8] I. Voitalov, P. van der Hoorn, R. van der Hofstad, and D. Krioukov, *Scale-free networks well done*, *Phys. Rev. Research* **1**, 033034 (2019).
- [9] J. Wilting and V. Priesemann, *25 years of criticality in neuroscience — established results, open controversies, novel concepts*, *Curr. Opin. Neurobiol.* **58**, 105–111 (2019).

Reviewer #2 (Remarks to the Author):

Universality, criticality and complexity of information propagation in social media

The authors analyze the time series of many hashtags across a number of social media datasets. They report findings of universal behavior, critical behavior across the different datasets, using arguments from statistical physics. They also use these findings to compare different propagation models, simple and complex contagion, to determine which may be more appropriate and when. While the paper is clearly written and the data are vast, I have significant qualms about the methods used and the conclusions drawn that prevent me from recommending the paper for publication.

My primary concern with this work is that the findings reported by the authors undermine the authors own claims. For instance, universality across their datasets is an important finding, but when I look at Fig 2, particularly 2c I see a clear failure of scaling collapse. I was especially surprised when first reading the paper that the authors found universal relations between social media-like datasets (all of which are essentially Twitter clones) and other datasets (stackoverflow and delicious). My surprise vanished when viewing 2c and seeing the latter groups exhibiting a clearly different scaling exponent. This leaves the social media-like datasets. It is not a major finding in my opinion that clones of Twitter exhibit dynamics similar to Twitter.

My secondary concern is how 'information' is used in the paper. The authors study hashtag dynamics. Presumably, this is related to information, but how? I know this sounds like an obvious/pointless remark, but deeper interrogation is needed, and I find the lack of clarity around this supposed connection to *universally damage* the credibility of these types of papers. Why does using a hashtag mean there is a spread of information? Can a hashtag be used without a corresponding information spread? Can information spread without a corresponding hashtag observation? These studies fail to ground their fundamental construct and fail to acknowledge the limitations of their construct's validity. Almost surely a more nuanced view of 'information' is needed, perhaps involving information theory but even then the mathematics of information can be quite divorced from social cognitive behavior.

Novelty. None of the statistical arguments the authors use are novel, instead coming from a base of work in statistical mechanics. Likewise, many studies have compared simple and complex contagion, including experiments that used social bots---something far more compelling than the analyses conducted here---and reporting findings that at least partially contradict the authors claims here.

Other concerns:

Logarithmic bins. Most analyses of the authors hinge on plots with linearly spaced points on a logarithmic scale. This point is not discussed in the manuscript. Logarithmic binning of data is well known to be problematic, and other approaches such as the ECDF should be used. (While cumulative distributions tend to carry along 'integration error' over the length of the curve, confidence intervals can be produced accounting for this). I think the authors will find that many of their pure power laws become far more mixed when the data are treated appropriately.

Poor treatment of model selection. The authors discuss log-likelihood ratio tests but also describe a process where models are picked by comparing p-values. I was surprised that AIC or BIC did not make an appearance. No discussion of model complexity, number of parameters, whether models are nested or not. Probably this is because the models the

authors study are not well suited to model selection. If this is the case, then the analysis, which hinges on model selection, is not sufficiently grounded. If this is not the case, the authors should justify and appropriately compare their models.

Use of literature. There are times when I question the appropriateness of cited literature. For instance, the authors write "there is large agreement on the fact that neuronal activity in the brain is universal and critical" followed by a number of references. Yet one of those references, ref. 23, states in their abstract:

- > This [criticality] hypothesis, however, is still controversial. Here we will
- > explain the concept of criticality and review the substantial objections to
- > the criticality hypothesis raised by skeptics. Points and counter points are
- > presented in dialog form.

Acknowledging that the reference is from 2012 and our understanding may have evolved in the interim, using that citation to support the authors' argument still reads to me as disingenuous.

Reviewer #3 (Remarks to the Author):

Universality, criticality and complexity of information propagation in social media

The paper addresses a highly relevant problem of peeking into the black-box of social dynamics to gain insight into the mechanisms behind information propagation on social media. With two "family" of models in mind - simple vs. complex contagion - the authors seek which is most relevant, given the observed propagation patterns. To achieve this they employ a statistical physics approach: first they extract observable distributions $P(T)$ and $P(S)$ for the duration and size of information cascades, then they use the observed distributions as a fingerprint by which to identify the system's hidden dynamics. Specifically, they find that both distributions asymptotically follow a power-law with exponents that can help distinguish between the different contagion mechanisms.

Using these observations, the authors find that the spreading dynamics of non-controversial issues (music, fashion etc.) exhibits a scaling consistent with simple contagion - a la disease spreading, in which a single exposure is sufficient to instigate a response. In contrast, more polemic issues (politics) spread via complex contagion, where multiple exposures are required for the contagion to propagate. I like the approach very much (perhaps since I myself have used it in the past - see below), and must congratulate the authors for being able to apply it on - the often elusive - social network data. The methodology is firm, and the results are convincing, corroborated across several data sources. I believe this paper has much potential to garner interest in the network science/complex system communities, and encourage publication in Nature Communications, pending my (few) comments below.

Neuronal dynamics. The paper, already in the abstract, and then throughout the introduction and the results section, consistently relates to neuronal dynamics as the example of the simple contagion class. Therefore, the reader expects to see a cross-disciplinary analysis, where, indeed, data from the two systems - social and brain - is analyzed and classified according to the two discussed universality classes. There are, however, two problems with setting this reference frame. First, I am not sure if neuronal excitations are, indeed, a broadly agreed upon example of simple contagion. There are multiple models in which neurons are assumed to follow threshold dynamics, where activation is a result of an integration over all neighbors. Perhaps I am missing something,

but if I understand the authors reference to neuronal dynamics correctly, I recommend avoiding this example, as it is, indeed, NOT the archetype of simple contagion.

The second point is less about the science, but rather about the narrative. I wonder if the reference to neuronal dynamics is at all needed. After all, the paper focuses solely on social contagion, and puts together a sufficient case in favor of the complex contagion scenario. No need for a straw-man. If the authors do wish to offer a simple (agreed upon) example of simple contagion then perhaps epidemic spreading is a better alternative. I feel that it, indeed, seems rather natural to ask whether information cascades in a social setting spread like viruses (simple) or via more complex propagation dynamics.

The technical details rest quiet firmly on Ref. 38, which the authors cite in several locations, e.g., to explain their measurement of avalanches, or how they extract the time resolution parameter Δt^* . It would improve the presentation if the authors provide more detail about the methodology of Ref. 38. If this is a digression, or harms the conciseness of the presentation, then perhaps a display item - text-box or illustrative figure - can do the job.

The authors compare social contagion to the random field Ising model (RFIM). The mapping seems reasonable: nodes negotiate their intrinsic tendency (temperature?), with the influence of their peers (network neighbors) and the general public sentiment (mean-field). I wonder if this mapping is just qualitative, or an actual proposition for a dynamic model that is consistent with the observed exponents. If the former, then the qualitative (i.e. not quantitative) nature of the analogy should be made clear. If the latter, however, I strongly recommend the authors to pursue this, and actually run the proposed model to show the emergence of the relevant universal exponent.

The above may benefit the paper in two directions: (i) Numerical simulations can show that, indeed, as the authors themselves state, many of the microscopic details do not affect the observed macro-scale patterns, strengthening the overall approach of the paper. This notion, a consensus among statistical physicists, is not always accepted by other disciplines, such as social sciences, that are within the potential readership of this work. (ii) Simulations may give rise to additional supporting predictions that can also be examined against the data. Indeed, knowing what to look for in the data is often crucial. Having said that, I do not wish to hijack the paper, and I therefore leave this comment to the discretion of the authors. As I indicated above, the paper is already strong in its present form, so it is up to the authors if they wish to pursue this additional support.

The distinction between the two types of information cascades, those dealing with light, vs. those propagating controversial issues should be further strengthened. This is, perhaps, the weak-point in the analysis. As the paper currently reads, it seems that the analyses detected two universality classes, which the authors then interpreted as topic based (using hashtags), and further analyzed as controversial vs. non-controversial (I presume, using their subjective association). If this is the case, it is more a hypothesis than a well-established result. Unless there is more to support this claim than what I picked up.

The paper uses many acronyms (BP,TS,RFIM etc.). Several times I had to look back to recall the meaning of a specific letter combination. As far as I remember, Nature Communications is rather generous in space. Hence I suggest to reduce abbreviations, and keep the presentation clean, except for self-expandatory ones (e.g., SF = scale-free).

Figure 2 - I think I am missing something. Panels (a) and (b) seem to me like $P(S)$ and $P(T)$, respectively. The Y-axis labels, however, seem to say otherwise. Why not just label the axes according to the functions presented. Also, in panel (d), it would seem a bar-plot is a more appropriate visualization, then the scattered symbols.

In my final comment, I am drawn to do something which I am slightly uncomfortable with: to refer the authors to a few of my own papers. I usually avoid doing this, however, I think in this case, you will see that they are highly relevant, both in terms of the methodology and in terms of the findings they offer. Specifically, while the authors discuss simple vs. complex contagion, there exists an alternative path by which to observe universal scaling relationships, and that is nonlinear interactions. Consequently, even simple contagion can yield a spectrum of potential scaling exponents if the nature of the contact mechanism is nonlinear. I think such mechanisms deserve attention in the introduction or discussion sections, as potential alternative pathways for observing distinct exponents. For example, these papers below link the interaction dynamics between the nodes to universal exponents of information propagation and cascades, indeed - highly relevant to the current paper's discussion:

- 1. Universality in network dynamics. Nature Physics 9, 673-681 (2013)**
- 2. Spatio-temporal propagation of signals in complex networks. Nature Physics 15, 403 (2019)**
- 3. Patterns of information flow in complex networks. Nature Communications 8, 2181 (2017)**

Finally, in this paper we offer an idea that is strongly related to the current contribution, linking the observed exponents to extract microscopic dynamic insight:

- 4. Constructing minimal models for complex system dynamics. Nature Communications 6, 7186 (2015)**

Now, I do not want to "force" the authors to cite my previous work. I leave it completely up to them to decide on its relevance and potential contribution to the discussion. Therefore, when receiving a revised version, I will decidedly disregard the authors response to this comment.

**I look forward to seeing this paper in the next round.
All my best,
Baruch Barzel**

First referee

This manuscript analyzes (hash)tags time series extracted from six online social media platforms and shows how cascades, when properly temporally circumscribed (parameter Δ^*), display a universal scaling in the distribution of their size (S) and duration (T), as well as in the relationship between their expected size ($\langle S \rangle$) given their duration.

I find the analysis sound, and the results are compelling and interesting. I particularly like how the results support a distinction between cascades on Twitter based on semantic, hinting that some themes cascade similarly to simple contagion, while other more controversial themes cascade with properties akin to complex contagion. I agree with the manuscript that such observation could have an important impact on our collective fight against the spread of misinformation on social media.

While the analysis and the novelty of the results are interesting enough to be granted publication in Nature Communications eventually, I believe that some work is still required to bring the manuscript up to the proper standards (see comments below).

We are grateful to the reviewer for the very positive report. The above paragraph provides a very good summary of our work.

For instance, I do not find the manuscript to be sufficiently self-contained in the sense that one needs to refer to the Supplementary Material on many occasions to understand the results presented in the main text. More details about the datasets, models and analysis should therefore be moved to the main text (or to the Material and Methods section).

As suggested by the reviewer, we appended a Methods section at the end of the revised version of the manuscript. There we provide a brief description of the data and the models, as well as a summary of the computational methods used in the analysis performed. Additional details about data and methods are still reported in the SM. We shortened the revised version of the SM to avoid repetitions of material already described in the Methods section.

I will be willing to reconsider the manuscript, once my comments listed below have been addressed.

We thank the reviewer for the willingness to reconsider our work. We dare to hope that our revisions will be sufficient to convince the referee. We provide below a point-to-point reply to all issues identified in the first round of review.

1. The manuscript says in its introduction that: **“there is large agreement on the fact that neuronal activity in the brain is universal and critical”**. After consulting with colleagues working in neuroscience, it seems that the concept of criticality in the brain is indeed quite popular among scientists with a physics-related background, but that it is still debated among scientists from the other realms of neuroscience. For instance, Ref. [13] says

This hypothesis is suggestive as it proposes that criticality could constitute a general and common organizing strategy in biology stemming from the physics of phase transitions. However, despite its implications, this is still in its infancy state as a well-founded theory and, as such, it has elicited some skepticism.

As another example, Ref. [6] says

Thus this hypothesis, and in particular the experimental work by Beggs and Plenz (2003), has triggered an avalanche of research, with thousands of studies referring to it. Nonetheless, experimental results are still contradictory.

While I believe that recent results like those of Ref. [5] are quite interesting, I nevertheless think the manuscript would be best served by depicting a more nuanced picture of the current state of the debate.

We thank the referee for this suggestion, which is similar to what recommended by all other reviewers. We followed the advise of the reviewer and changed the narrative of the manuscript by avoiding the use of neuronal activity as an example of a critical and universal dynamical process. Neuronal avalanches are only mentioned in reference to our own analysis, shown in SM D, to state that our methodology provides results that are consistent with previous studies.

2. (neuronal activity vs simple contagion): The manuscript should expand on how the observation that the critical exponent values measured for neuronal activity **“are those of the universality class of the mean-field branching process”** and therefore that neurons **“influence each other according to a simple contagion process”** compare with temporal and spatial summation mechanisms^a which are reminiscent of a threshold model and thus of complex contagion. Indeed, it is my understanding that the universality observed in scaling analysis usually comes at the price of not ignoring the “microscopic” details of the dynamics [10]. Hence, it is one thing to say that neuronal activity belongs to the same universality class as BP, but it is another to say that **“neuronal activity may be modeled as a simple contagion process [. . .]”**.

^a[https://en.wikipedia.org/wiki/Summation_\(neurophysiology\)](https://en.wikipedia.org/wiki/Summation_(neurophysiology))

As stated above, we realized that the example of neuronal avalanches as a generally accepted system exhibiting simple contagion dynamics is not sufficiently solid and hence we removed it.

3. (Datasets, BP and RFIM): Elements from sections A, B, D, E anf F should find their way in the Material and Methods section of the manuscript.

We agree with the referee. We created a new section Methods where we summarize material previously appearing in the SM.

4. (Deviation from RFIM): On page 3, the manuscript says

At the aggregate level, each social media displays a critical behavior that is compatible with the Random Field Ising Model, indicating that, plausibly, information propagates in social media according to a complex contagion process.

but later says

There is an apparent mismatch between our estimates $\hat{\alpha}$ and $\hat{\gamma}$ and the RFIM predictions $\alpha = 7/2$ and $\gamma = 2$. The mismatch can be theoretically explained by the peculiar shape of the scaling function characterizing the distribution of avalanche duration, which affects also the estimate of $\hat{\gamma}$ [52]. Difficulties in observing the asymptotic exponents of the RFIM due to the effect of the scaling functions emerge also in numerical simulations of the RFIM and are well known [19].

Looking at these two references [4, 10], it was not at all clear to me why $\hat{\alpha} = 5/2$ shown on Fig. 2 should be considered as coherent with Random Field Ising Model for which $\alpha = 7/2$ (and consequently for $\hat{\gamma} = 6/5$ and $\gamma = 2$). This justification should be better explained and illustrated.

We thank the referee for this consideration, as it pushed us to significantly improve our analysis.

We realized we offered a complicated explanation of the phenomenon. Briefly, the mismatch between empirically measured and theoretically predicted values of the critical exponents of the RFIM is due to finite size effects. The issue has been studied for the low-dimensional version of the RFIM, see Ref. [10].

In the revised version of the paper, we bypass the finite size issue by measuring the maximum likelihood estimators $\hat{\tau}$ and $\hat{\alpha}$ from simulations of the models. For BP, we do not find significant effects due to finite size: the exponents measured from numerical simulations well approximate those derived from the the mean-field theory and valid in the thermodynamic limit. For RFIM the estimator of τ is similar to the one valid in the limit of large system sizes; the estimator of α instead displays a deviation from the value expected in the thermodynamic limit. On the other hand, we note that both the critical exponents of the RFIM estimated from numerical simulations of the model are compatible with those obtained on the empirical data sets.

We changed the manuscript to simplify the discussion about the effect of the finite size of the system for the estimation of the critical exponents. In particular, we generated a new version of Figure 2 where the distributions obtained from the empirical data sets are compared with those obtained via numerical simulations of both BP and RFIM on finite size systems.

5. (Figure 1): The meaning of “percolation strength” is not straightforward in this context. I understand that the manuscript adopts the approach presented in Ref. [8] (from the same authors) but for the sake of completeness, and because the approach is not standard yet, the clarity of the manuscript would greatly improve if a description of the method (including the role of the susceptibility) were briefly explained in a Material and Methods section. Section D of the SM could also be moved to the manuscript, but I believe a shorter note would be sufficient.

The Methods section now contains a brief introduction to the approach presented in Ref. [8], including a definition of percolation strength and susceptibility, how they are computed and the role they play in our analysis.

6. (Figure 2): The scalings obtained for BP and RFIM should appear on every subplot (A, B and C), and the scaling found using the method from Ref. [3] should be shown using another line style and color.

We updated the visualization of Figure 2. We do not show best fits obtained with the method of Ref. [3] for each empirical distribution. We found that adding many lines to the plots would make the figure too dense and difficult to read. We note that values of the best estimators of each data set are already shown in Figure 2d. Also, we have added a table in SM C where we report the numerical values of the estimators.

7. (natural resolution): The manuscript states that “[. . .] Δ^* representing the natural resolution for observing information avalanches.” Do the authors have thought about possible explanations for the substantial variation of these time scales from one dataset to the other (i.e. “from $\Delta^* 1500$ s for Twitter to $\Delta^* = 30000$ s for Telegram”).

Our work in Ref. [8] suggests that the optimal resolution Δ^* decreases as the rate of the corresponding process increases. This finding is confirmed in Fig R1 where we show the optimal resolution Δ^* as a function of the average daily rate of activity in each platform.

8. (compatibility with mean-field model): It would be instructive if the manuscript could expand more on the claim that

The compatibility of avalanche statistics with those of a homogeneous mean-field model is not surprising given that in some social media there is no underlying network among users and in the others there are mechanisms for the propagation of information that bypass it.

by providing concrete examples. For instance, what mechanism allows the cascade of a hashtag to bypass the underlying network structure of Telegram?

We thank the referee for making this comment. In the revised version of the manuscript, we included some explicit examples of why we believe that a mean-field description of our systems is appropriate. We repeat these examples here.

Figure R1: **Relation between optimal resolution Δ^* and average rate.** We show the optimal resolution Δ^* against the average daily rate in the six online social media considered in our study.

Telegram is a messaging platform and the data collected in Ref. [1] are “channel-based.” Telegram channels are “one-to-many communication” systems and “can be created by any Telegram user, and other Telegram users can join or subscribe to the channel to read its content” [1]. Hence, Telegram channels are a perfect example of a mean-field system, where every user in the channel communicates with every other user in that channel.

In Stackoverflow, there is no underlying network. Users access content using common sets of tools provided by the platform, i.e., search bar, “Top question” section.

In Twitter, there is an underlying network. A post could be seen by followers of the user who made the Tweet. However, this is not the only mechanism that exposes a user to a Tweet. Trending Tweets can appear in a user’s feed, as well as Tweets made by a second neighbor and Tweets that the platform believes of interest for the user. In general, the platform manages to increase as much as possible the engagement of the users, effectively bypassing the follower-followee network.

9. (homogeneous mean-field model): It is not clear from the manuscript alone why the RFIM is a “homogeneous mean field model”.

The Random Field Ising Model is not necessarily a homogeneous mean-field model. In principle, it can be defined on top of an arbitrary network. We are considering it in its mean-field version, i.e., on top of a complete network. Our choice is motivated by the arguments used to address the above criticism.

10. (fitting power laws): It would be instructive if the authors explained why they used the methods of Ref. [3] instead of the more recent approaches proposed in Refs. [2, 11].

When inferring the best estimator of the exponents of the avalanche distribution $P(S)$ and $P(T)$, we

simply compute the maximum likelihood estimators and the associated errors. This is a standard procedure and, as such, also used in Ref. [3].

We relied on Ref. [3] to develop our fitting technique. The method of Ref. [2] is the same of that in Ref. [3]. We also found that the methods in Ref. [11] are inappropriate for the application at hand. In Ref. [11] the authors consider the class of “regularly varying distributions,” i.e., distributions “allowing for deviations from pure power laws by means of a slowly varying function.” In our case, however, the distributions we fit deviate quickly from the power-law behaviour when the models are far from their critical points. We note that sufficiently far from criticality the distributions characterizing the models are exponential, thus making the method of Ref. [11] not suitable for our case.

11. Why did the authors considered “avalanches that contain at least two avalanches larger than S_{min} ”. Is the method robust if they chose other values (including 1) instead of 2?

We thank the referee for the question. We do not require “at least two avalanches with $S \geq S_{min}$ ” but actually at least two avalanches with $S \geq S_{min}$ AND with different size. This is necessary because we need at least two non-zero values of $P(S)$ to make the fit. We have expanded the text of the revised version of the paper to clarify this point. Results are also robust if the number of non-zero values of P is chosen to be different from 2. In Fig. R2 we show the classification results if at least 10 non-zero values are required, where we have used $S_{min} = 10$ and have set the threshold over the p -values of the individual fits to 0.1, as in the main text.

12. The manuscript says “[. . .] is subject to the social influence exerted by the agents she interacts with [. . .]” when, in the RFIM, the agent interacts with everyone. Does this difference, from a network to a mean-field approach, make a difference?

In general, it does. As we replied above, we adopt a mean-field version of the model for simplicity and because it represents a good choice for the description of information avalanches in social media that do not rely on an underlying social network.

13. (Equation S4): Equation (S4) should read

$$\hat{\eta} = 1 + Z \left(\sum_{i=1}^Z \ln \frac{x_i}{x_{min}} \right)^{-1}$$

We thank the referee for this observation. We updated the manuscript.

14. (Introduction): I would suggest incorporating the footnote into the main text. Otherwise the first paragraph feels somewhat superficial.

We thank the referee for this observation. We updated the manuscript.

Figure R2: **Robustness against the minimal number of non-zero values of $P(S)$ for the time series to be classified.** We require here a time series to have at least 10 different values of the avalanche size that are $\geq S_{min}$. a) Probability that the log-likelihood ratio test favors the Random Field Ising Model over Branching Process (red), or vice versa (teal), using a threshold 0.1 over the p -values. Only time series that are sufficiently well fitted by both models are considered in the analysis, see panel b. Error bars represent σ/N , where N is the sample size and $\sigma = \sqrt{0.25N}$ is the standard deviation of a binomial distribution with probability of success equal to 1/2. Asterisks are used to denote significant deviations from the unbiased binomial model, i.e., two asterisks indicate for $p < 0.01$ and one asterisk stands for $p < 0.1$. b) We report the fraction of time series that are classified in the Random Field Ising Model class (red), the fraction of time series that are classified as branching processes (teal), the fraction of time series that is classified as neither (purple) and the fraction of time series that pass both statistical tests (yellow). In this case, the log-likelihood ratio test is required for model selection, see panel a. Here we set to 0.1 the threshold over the p -values.

15. (Page 4): Something is missing in “are so basic that underlie information diffusion”.

We thank the referee for this observation. We expanded the sentence to better clarify the point.

16. (Page 4): What “topological features” are the authors referring to here?

We were referring to the characteristics of the social network of users over which messages (carrying specific hashtags) spread. Indeed, for the specific problem of predicting future popularity given early observations, the features of the network could play an important role [12]. We changed the sentence to better explain what we meant.

17. (Code on Github): Have the authors considered archiving the repository containing the Twitter data using services like Zenodo^a to ensure its long-term availability?

^a<https://guides.github.com/activities/citable-code/>

We updated the Twitter data on Zenodo. We further uploaded all the codes needed to reproduce the analysis and a notebook where the analysis is actually reproduced on the twitter dataset. This material is available at [7].

18. (Figures 1, 2 and 3): Please add the symbol associated with the quantity whenever possible. For instance, “Size”, “Rescaled duration” and “Disorder” should read “Size S ”, “Rescaled duration (T/Δ^*)” and “Disorder (R)”.

We changed the layout of our figures. We now use mathematical expressions in the axis labels of the plots.

19. (Figure 2): Would it be useful to display the raw distributions $P(S)$ and $P(T)$ in an inset?

Logarithmic binning is used only for visualization purposes. The scaling exponents are obtained from raw data. Distributions obtained via linear binning are very similar to those obtained using logarithmic binning, but they have noisier tails. For completeness, we added the cumulative distribution functions of S and T to SM E.

20. (Figure 3C): The color used for BP [RGB: (0, 139, 139)] is closer to teal [RGB: (0, 128, 128)] than to blue.

We thank the referee for this observation. We updated the manuscript.

21. (Figure 4): A table or an histogram would be a better choice than a word cloud where the relative rank between word is biased by their length.

We have added section K to the SM, where we report the hashtags shown in the word cloud, ranked according to their popularity.

22. (Bibliography): Adding the title of the references (and a clickable link) would greatly help the reader to follow the literature the manuscript builds upon. If using the revtex document class, simply add the option longbibliography and include the doi field in the .bib file. The package hyperref may also be required.

Thanks for the suggestion. We change the latex template of the manuscript and included the titles of all referenced papers. We also included the DOIs of the papers.

Second referee

The authors analyze the time series of many hashtags across a number of social media datasets. They report findings of universal behavior, critical behavior across the different datasets, using arguments from statistical physics. They also use these findings to compare different propagation models, simple and complex contagion, to determine which may be more appropriate and when. While the paper is clearly written and the data are vast, I have significant qualms about the methods used and the conclusions drawn that prevent me from recommending the paper for publication.

We thank the reviewer for the time dedicated to our manuscript. We appreciate the criticisms as they stimulated us to improve some technical parts of the analysis and the overall narrative of the paper. We hope that the reviewer will find the revised version of the manuscript suitable for publication.

1. My primary concern with this work is that the findings reported by the authors undermine the authors own claims. For instance, universality across their datasets is an important finding, but when I look at Fig 2, particularly 2c I see a clear failure of scaling collapse. I was especially surprised when first reading the paper that the authors found universal relations between social media-like datasets (all of which are essentially Twitter clones) and other datasets (stackoverflow and delicious). My surprise vanished when viewing 2c and seeing the latter groups exhibiting a clearly different scaling exponent. This leaves the social media-like datasets. It is not a major finding in my opinion that clones of Twitter exhibit dynamics similar to Twitter.

We appreciate this criticism as it motivated us to give further evidence in support of our claim for universal behavior. In the original version of the manuscript the claim was based on the similarity of the exponent values for the power-law scalings of the various data sets (i.e., Figure 2d). Following the reviewer suggestion we now produce the data collapse by rescaling variables by their respective average values. We performed this operation in the revised version of Figure 2, see also Figure R2 in this reply.

The linear fit of the curves in Fig. 2c provides with the estimator $\hat{\gamma}$, shown in Fig. 2d. The results obtained for StackOverflow ($\hat{\gamma} = 1.31 \pm 0.02$) and Delicious ($\hat{\gamma} = 1.25 \pm 0.01$) are rather close to those obtained for Telegram ($\hat{\gamma} = 1.31 \pm 0.03$) and Weibo ($\hat{\gamma} = 1.26 \pm 0.05$).

Concerning the distinction between Twitter clones *vs.* Stackoverflow/Delicious made by the reviewer, although some degree of resemblance exists between platforms, they differ in significant aspects from a user perspective. For example, in Twitter there is an underlying social network that relates followers to followees. In Telegram and Stackoverflow, there is no such a network and discussions are channel/topic based.

The apparent universality across platforms indeed indicates that there are individual characteristics, potentially relevant for the diffusion of posts, that have little effect on the shape of the distributions of interest. This is one of the main messages of our multi-platform analysis which we hope is now clearly communicated by the revised version of Figure 2.

2. My secondary concern is how 'information' is used in the paper. The authors study hashtag dynamics. Presumably, this is related to information, but how? I know this sounds like an obvious/pointless remark, but deeper interrogation is needed, and I find the lack of clarity around this supposed connection to *universally damage* the credibility of these types of papers. Why does using a hashtag mean there is a spread of information? Can a hashtag be used without a corresponding information spread? Can information spread without a corresponding hashtag observation? These studies fail to ground their fundamental construct and fail to acknowledge the limitations of their construct's validity. Almost surely a more nuanced view of 'information' is needed, perhaps involving information theory but even then the mathematics of information can be quite divorced from social cognitive behavior.

We acknowledge that we study (hash)tag dynamics, and we understand that the reviewer is concerned about the use of the word *information* in this context. Indeed information can be an ambiguous expression used with very different meaning in different context, both in natural and technical languages. Unfortunately it is difficult to come up with an effective synonym in this contexts. We hope that the revised version of the paper clearly conveys that we are concerned with tag diffusion.

3. Novelty. None of the statistical arguments the authors use are novel, instead coming from a base of work in statistical mechanics. Likewise, many studies have compared simple and complex contagion, including experiments that used social bots—something far more compelling than the analyses conducted here—and reporting findings that at least partially contradict the authors claims here.

In lack of specific references we cannot be sure which “more compelling studies reported findings that at least partially contradict the authors claims here.” the reviewer is referring to. There certainly exist studies that approach the distinction between complex and simple contagion by focusing on the microscopic mechanisms of contagion. Our approach is complementary to those. We uncover what are the large-scale consequences (in terms of avalanches of information spreading) of the two types of contagion. In this respect, where complex contagion is concerned we believe our results are original.

As for our statistical arguments, we indeed make no strong claim of novelty. The focus here is the characterization of avalanches and their theoretical interpretation in terms of simple dynamical models. We do claim some element of methodological novelty, though. The first consists in the selection of the temporal resolution Δ^* based on the novel technique developed in Ref. [8]. The current work is the first that applies such a technique to the analysis of empirical data. Indeed we also considered data that are not related to our main focus, i.e., social media, to support the claim that the technique is reliable and produces robust results regardless of the system at hand. Novel techniques are used also in the second part of our work, where we generalize the methods of Ref. [3] to discriminate between simple and complex contagion. Indeed, as we explicitly stated in the main text, the methods of Ref. [3] are suitable to discriminate between two candidate probability distributions. Here, instead, we re-adapt the technique to discriminate between two different dynamical models.

4. Logarithmic bins. Most analyses of the authors hinge on plots with linearly spaced points on a logarithmic scale. This point is not discussed in the manuscript. Logarithmic binning of data is well known to be problematic, and other approaches such as the ECDF should be used. (While cumulative distributions tend to carry along 'integration error' over the length of the curve, confidence intervals can be produced accounting for this). I think the authors will find that many of their pure power laws become far more mixed when the data are treated appropriately.

Logarithmic binning is used for visualization purposes only. The power-law exponents of the distributions in Figure 2 are computed using maximum likelihood estimation, which uses raw data and does not rely on a binning procedure. Also the fit of the individual time series with the dynamical models is based on maximum likelihood estimation and also in this case does not rely on the binning of the data.

We agree with the reviewer that the visualizations of Figure 2 may look different if the binned distributions are replaced by complementary cumulative distribution functions (CCDFs). We added CCDF visualizations to the SM, section E. For convenience, we report them also here, including confidence intervals. Confidence intervals appear very large on the logarithmic scale. The CCDFs denote good data collapse. Further, a large portion of the curves appear straight on the log-log plot, thus compatible with a power-law scaling.

Figure R3: **Empirical Complementary Cumulative Distribution Function of avalanche size and duration.** We show the Empirical Complementary Cumulative Distribution Function of avalanche size (panel A) and duration (panel B) for the six social media considered in our manuscript. Each curve is accompanied by a color-filled area that marks the width of the confidence intervals. The confidence interval represents the true CCDF with probability 0.99. To achieve data collapse, data in panel a are shown as a function of $S/\langle S \rangle$ rather than as a function of S and data in panel b are shown as a function of $T/\langle T \rangle$ rather than as a function of T .

5. Poor treatment of model selection. The authors discuss log-likelihood ratio tests but also describe a process where models are picked by comparing p-values. I was surprised that AIC or BIC did not make an appearance. No discussion of model complexity, number of parameters, whether models are nested or not. Probably this is because the models the authors study are not well suited to model selection. If this is the case, then the analysis, which hinges on model selection, is not sufficiently grounded. If this is not the case, the authors should justify and appropriately compare their models.

We apologize for the lack of clarity. In the original submission, many details were presented in the SM only. In the revised submission, these details are now summarized in the section Methods appearing at the end of the paper. We believe that this modification has improved the readability of the paper, and solved the issue raised by the referee. For completeness, we reply to the various criticisms below.

Let us first briefly recapitulate our strategy. In our analysis we follow the same pipeline as of Ref. [3]. The p -value quantifies the compatibility of an individual time series with respect to an individual model, i.e., RFIM or BP. If the p -value is below a certain level of statistical significance, the fit is not good enough and the hypothesis that the data were generated according to that model must be rejected. Hence, we compute the p -value of both our fits and reject a model every time its p -value is not large enough. If one model is rejected and the other is not, the non-rejected model is the selected one, as prescribed in Ref. [3]. If both models are rejected, no model is selected, as explicitly stated in our work. If both models are not rejected, we perform model selection via the likelihood ratio test, as prescribed in Ref. [3].

About the number of parameters: both models have one parameter only. We apologize for not stressing enough this point. This should be now clear from the description of the models in the Methods section. We stress this fact in the main text too, mentioning that for the branching process “the only tunable parameter of the model is denoted by n ,” and that “ R is the only tunable parameter of the” Random Field Ising Model.

Finally, we note that the branching process and the Random Field Ising Model are not nested. This fact should be more transparent now that we have introduced the Methods section (previously SM E, F). To defend our case, let us also stress some of the important differences between RFIM and BP. The models have two completely different dynamics. The Random Field Ising Model is an Hamiltonian model that takes place on a network, and its dynamics is driven by an external force. RFIM reaches a stationary state with certainty, when $H = +\infty$ and all nodes are flipped. The branching process is not Hamiltonian, does not takes place on top of a network, is self-organized rather than driven, and is not guaranteed to reach a stationary state. The models do not even undergo the same type of phase transition (the Random Field Ising Model is depinning, the branching process is absorbing).

In terms of computational complexity, a realization of the branching process is linear in the size of the avalanche, while a realization of the Random Field Ising Model scales as $N \log N$, with N equals to the system size, because the N values of the random fields h_i must be sorted.

The referee raises an important point about the suitability of the two models for model selection. Our contention is that they are. The techniques developed in Ref. [3] are suitable to (i) fit and (ii) compare the fits of probability distributions, possibly with a fat tail. In particular, the authors

consider power laws, exponential distributions and power laws with exponential cutoffs, the three distributions relevant to the present study. Moreover, we test our fitting technique on the two models in SM I and we verify that our method is able to retrieve the ground truth in synthetic data. Finally, the models are compared appropriately. The referee suggests the use AIC or BIC, but these two criteria are identical to the ratio of the likelihoods if the number of parameters of the models to be compared is the same. This is exactly our situation, where both the branching process and the Random Field Ising Model have one tunable parameter only.

6. Use of literature. There are times when I question the appropriateness of cited literature. For instance, the authors write "there is large agreement on the fact that neuronal activity in the brain is universal and critical" followed by a number of references. Yet one of those references, ref. 23, states in their abstract:

This [criticality] hypothesis, however, is still controversial. Here we will explain the concept of criticality and review the substantial objections to the criticality hypothesis raised by skeptics. Points and counter points are presented in dialog form.

Acknowledging that the reference is from 2012 and our understanding may have evolved in the interim, using that citation to support the authors' argument still reads to me as disingenuous.

We acknowledge that we have used literature from neuroscience without fully understanding how intricate and still debated the hypothesis of universality/criticality in the brain is. We thank the referee for this observation. We decided to change the narrative of the manuscript, removing any claim about neuronal dynamics. These claims were indeed not central for the content of the paper, so none of our results has been affected by this change.

Third referee

The paper addresses a highly relevant problem of peeking into the black-box of social dynamics to gain insight into the mechanisms behind information propagation on social media. With two "family" of models in mind - simple vs. complex contagion - the authors seek which is most relevant, given the observed propagation patterns. To achieve this they employ a statistical physics approach: first they extract observable distributions $P(T)$ and $P(S)$ for the duration and size of information cascades, then they use the observed distributions as a fingerprint by which to identify the system's hidden dynamics. Specifically, they find that both distributions asymptotically follow a power-law with exponents that can help distinguish between the different contagion mechanisms.

Using these observations, the authors find that the spreading dynamics of non-controversial issues (music, fashion etc.) exhibits a scaling consistent with simple contagion - a la disease spreading, in which a single exposure is sufficient to instigate an response. In contrast, more polemic issues (politics) spread via complex contagion, where multiple exposures are required for the contagion to propagate. I like the approach very much (perhaps since I myself have used it in the past - see below), and must congratulate the authors for being able to apply it on - the often elusive - social network data. The methodology is firm, and the results are convincing, corroborated across several data sources.

I believe this paper has much potential to garner interest in the network science/complex system communities, and encourage publication in Nature Communications, pending my (few) comments below.

We thank the referee for the very positive comments about our manuscript. We followed the suggestions by the reviewer and revised the manuscript accordingly. In the following, we provide a point-to-point reply to all issues raised by the referee. We hope the reviewer will recommend our manuscript for publication.

1. Neuronal dynamics. The paper, already in the abstract, and then throughout the introduction and the results section, consistently relates to neuronal dynamics as the example of the simple contagion class. Therefore, the reader expects to see a cross-disciplinary analysis, where, indeed, data from the two systems - social and brain - is analyzed and classified according to the two discussed universality classes. There are, however, two problems with setting this reference frame. First, I am not sure if neuronal excitations are, indeed, a broadly agreed upon example of simple contagion. There are multiple models in which neurons are assumed to follow threshold dynamics, where activation is a result of an integration over all neighbors. Perhaps I am missing something, but if I understand the authors reference to neuronal dynamics correctly, I recommend avoiding this example, as it is, indeed, NOT the archetype of simple contagion.

We thank the reviewer for this comment. As suggested by the referee, we revised the narrative

of the paper by avoiding unnecessary statements about the statistical properties that characterize neuronal avalanches in the brain.

2. The second point is less about the science, but rather about the narrative. I wonder if the reference to neuronal dynamics is at all needed. After all, the paper focuses solely on social contagion, and puts together a sufficient case in favor of the complex contagion scenario. No need for a straw-man. If the authors do wish to offer a simple (agreed upon) example of simple contagion then perhaps epidemic spreading is a better alternative. I feel that it, indeed, seems rather natural to ask whether information cascades in a social setting spread like viruses (simple) or via more complex propagation dynamics.

As we already mentioned, we changed the narrative of the paper following the advise by the reviewer.

3. The technical details rest quiet firmly on Ref. [8], which the authors cite in several locations, e.g., to explain their measurement of avalanches, or how they extract the time resolution parameter Δ^* . It would improve the presentation if the authors provide more detail about the methodology of Ref. [8]. If this is a digression, or harms the conciseness of the presentation, then perhaps a display item - text-box or illustrative figure - can do the job.

We have added the section Methods to the main manuscript and the approach of Ref. [8] is now part of it, rather than a section of the SM.

4. The authors compare social contagion to the random field Ising model (RFIM). The mapping seems reasonable: nodes negotiate their intrinsic tendency (temperature?), with the influence of their peers (network neighbors) and the general public sentiment (mean-field). I wonder if this mapping is just qualitative, or an actual proposition for a dynamic model that is consistent with the observed exponents. If the former, then the qualitative (i.e. not quantitative) nature of the analogy should be made clear. If the latter, however, I strongly recommend the authors to pursue this, and actually run the proposed model to show the emergence of the relevant universal exponent. The above may benefit the paper in two directions: (i) Numerical simulations can show that, indeed, as the authors themselves state, many of the microscopic details do not affect the observed macro-scale patters, strengthening the overall approach of the paper. This notion, a consensus among statistical physicists, is not always accepted by other disciplines, such as social sciences, that are within the potential readership of this work. (ii) Simulations may give rise to additional supporting predictions that can also be examined against the data. Indeed, knowing what to look for in the data is often crucial. Having said that, I do not wish to hijack the paper, and I therefore leave this comment to the discretion of the authors. As I indicated above, the paper is already strong in its present form, so it is up to the authors if they wish to pursue this additional support.

We did consider numerical simulations of the models in the previous version of the manuscript. For example, results of numerical simulations of the Random Field Ising Model were reported in the SM. Also, individual time series are fitted against results obtained by numerically simulating the Branching Process and the Random Field Ising Model.

However, we did not yet fully exploit the utility of numerical simulations of the models to predict the scaling of the distributions of Figure 2. In the revised version of the manuscript, we use numerical simulations of both models to compute the maximum likelihood estimators of the exponents τ and α and compare the empirical results with these quantities to assess whether the Random Field Ising Model better explains the data. We thank the reviewer for recommending us to perform this quantitative comparison between empirical data and models.

5. The distinction between the two types of information cascades, those dealing with light, vs. those propagating controversial issues should be further strengthened. This is, perhaps, the weak-point in the analysis. As the paper currently reads, it seems that the analyses detected two universality classes, which the authors then interpreted as topic based (using hashtags), and further analyzed as controversial vs. non-controversial (I presume, using their subjective association). If this is the case, it is more a hypothesis than a well-established result. Unless there is more to support this claim than what I picked up.

We agree with the reviewer that the map between dynamic universality classes and the topic of hashtags is more a speculation than an actual result of the paper. We remark that our speculation is supported by a quantitative analysis of Twitter data performed by Romero and Kleinberg [9]. Also, it appears clear that certain hashtags that we classify in the RFIM class are effectively related to political movements or protests (e.g., ChileDespierta, HumanRightsandDemocracyAct), while certain hashtags in the BP class regard conversational topics (e.g., RWC2019, GOT7_CallMyName). Validating our speculation would require a semantic analysis of the hashtags that is outside the main purpose of this paper, and well beyond our expertise.

To address the concern by the reviewer, we slightly modified the text of the paper to further stress that ours is mainly a speculation rather a result.

6. The paper uses many acronyms (BP,TS,RFIM etc.). Several times I had to look back to recall the meaning of a specific letter combination. As far as I remember, Nature Communications is rather generous in space. Hence I suggest to reduce abbreviations, and keep the presentation clean, except for self-expandatory ones (e.g., SF = scale-free).

We removed the acronym TS for time series from the manuscript and from the SM. However, we would like to keep the acronyms RFIM and BP for the Random Field Ising Model and the for the branching process respectively. This is because these acronyms appear often and sometimes they appear many times in close sentences, see for example the caption of Fig. 2.

7. Figure 2 - I think I am missing something. Panels (a) and (b) seem to me like $P(S)$ and $P(T)$, respectively. The Y-axis labels, however, seem to say otherwise. Why not just label the axes according to the functions presented. Also, in panel (d), it would seem a bar-plot is a more appropriate visualization, then the scattered symbols.

We have changed the labels of the figures. Specifically, now the axes of Figure 2 are labeled according to the functions presented.

8. In my final comment, I am drawn to do something which I am slightly uncomfortable with: to refer the authors to a few of my own papers. I usually avoid doing this, however, I think in this case, you will see that they are highly relevant, both in terms of the methodology and in terms of the findings they offer. Specifically, while the authors discuss simple vs. complex contagion, there exists an alternative path by which to observe universal scaling relationships, and that is nonlinear interactions. Consequently, even simple contagion can yield a spectrum of potential scaling exponents if the nature of the contact mechanism is nonlinear. I think such mechanisms deserve attention in the introduction or discussion sections, as potential alternative pathways for observing distinct exponents. For example, these papers below link the interaction dynamics between the nodes to universal exponents of information propagation and cascades, indeed - highly relevant to the current paper's discussion:

1. Universality in network dynamics. *Nature Physics* 9, 673-681 (2013)
2. Spatio-temporal propagation of signals in complex networks. *Nature Physics* 15, 403 (2019)
3. Patterns of information flow in complex networks. *Nature Communications* 8, 2181 (2017)

Finally, in this paper we offer an idea that is strongly related to the current contribution, linking the observed exponents to extract microscopic dynamic insight:

4. Constructing minimal models for complex system dynamics. *Nature Communications* 6, 7186 (2015)

Now, I do not want to "force" the authors to cite my previous work. I leave it completely up to them to decide on its relevance and potential contribution to the discussion. Therefore, when receiving a revised version, I will decidedly disregard the authors response to this comment.

We thank the referee for pointing out these interesting works, which indeed we know well. Despite they are conceptually relevant for our work, as they deal with inferring and modelling universality of dynamical processes that run on top of complex networks, we are unfortunately not in the conditions to apply such a framework to our data for lack of knowledge of the actual social network underlying the diffusion of information. Still, we thank the reviewer for indicating an interesting direction for future research.

We included a few sentences discussing this possibility in the conclusion of the paper, and we added the references to the papers indicated by the reviewer.

List of changes

1. Following the suggestions by the reviewers, we substantially changed the narrative of the manuscript. All the changes are highlighted in red to ease their identification.
2. We moved content from the SM to the Methods section appearing at the end of the revised version of the manuscript.
3. We updated Fig. 2, and we modified the axis labels in all figures.
4. The maximum likelihood estimators of the avalanche exponents obtained from the data are now compared against the same estimators obtained from numerical simulations of the models.
5. Code and data have been made publicly available at <https://zenodo.org/record/5779063#.YbhyBH3P1Yg>.
6. We added content to the SM. Changes are highlighted in red. Also, we added Tables S7 and S8, Figure S5.
7. We revised the bibliography by including the title and the DOI of each reference.

Bibliography

- [1] Jason Baumgartner, Savvas Zannettou, Megan Squire, and Jeremy Blackburn. The pushshift telegram dataset. *arXiv preprint arXiv:2001.08438*, 2020.
- [2] Anna D Broido and Aaron Clauset. Scale-free networks are rare. *Nature communications*, 10(1):1–10, 2019.
- [3] Aaron Clauset, Cosma Rohilla Shalizi, and Mark EJ Newman. Power-law distributions in empirical data. *SIAM review*, 51(4):661–703, 2009.
- [4] Serena di Santo, Pablo Villegas, Raffaella Burioni, and Miguel A Muñoz. Simple unified view of branching process statistics: Random walks in balanced logarithmic potentials. *Physical Review E*, 95(3):032115, 2017.
- [5] Leandro J Fosque, Rashid V Williams-García, John M Beggs, and Gerardo Ortiz. Evidence for quasicritical brain dynamics. *Physical Review Letters*, 126(9):098101, 2021.
- [6] Miguel A Munoz. Colloquium: Criticality and dynamical scaling in living systems. *Reviews of Modern Physics*, 90(3):031001, 2018.
- [7] Daniele Notarmuzi, Claudio Castellano, Alessandro Flammini, Dario Mazzilli, and Filippo Radicchi. <https://zenodo.org/record/5779063#.Ybhyi33P1Yg>, 2021.
- [8] Daniele Notarmuzi, Claudio Castellano, Alessandro Flammini, Dario Mazzilli, and Filippo Radicchi. Percolation theory of self-exciting temporal processes. *Physical Review E*, 103(2):L020302, 2021.
- [9] Daniel M Romero, Brendan Meeder, and Jon Kleinberg. Differences in the mechanics of information diffusion across topics: idioms, political hashtags, and complex contagion on twitter. In *Proceedings of the 20th international conference on World wide web*, pages 695–704, 2011.
- [10] James P Sethna, Karin A Dahmen, and Christopher R Myers. Crackling noise. *Nature*, 410(6825):242–250, 2001.
- [11] Ivan Voitalov, Pim van der Hoorn, Remco van der Hofstad, and Dmitri Krioukov. Scale-free networks well done. *Physical Review Research*, 1(3):033034, 2019.

- [12] Lilian Weng, Filippo Menczer, and Yong-Yeol Ahn. Predicting successful memes using network and community structure. In *Proceedings of the International AAAI Conference on Web and Social Media*, volume 8, 2014.
- [13] Jens Wilting and Viola Priesemann. 25 years of criticality in neuroscience—established results, open controversies, novel concepts. *Current opinion in neurobiology*, 58:105–111, 2019.

REVIEWERS' COMMENTS

Reviewer #1 (Remarks to the Author):

I am satisfied by the authors' answers to my comments and to the modifications they made to the manuscript.

My recommendation is to accept the manuscript for publication once the second reviewer, whose background on the topic appears to be deeper than mine, is satisfied.

Reviewer #2 (Remarks to the Author):

I have considered the revised manuscript. I appreciate the efforts to clarify their paper. However, I still struggle to see the importance of the study-peeling back all the statistical physics, the authors seem to be investigating simple versus complex contagion and reporting "sometimes it's both". Why is this impactful? Likewise, why is it impactful to find that social systems are distinct from earthquakes and rat brains? Despite the rebuttal, I remain skeptical of some arguments employed by the authors. Finally, I find the paper exceedingly hard to read, harder than before. This alone in my opinion makes it unsuitable for a general audience venue such as Nature Communications.

In the discussion, the authors write,

> Also, in contrast with the vast majority of previous studies where purely diffusive models have been considered [37], we showed that information propagation in social media is often better described by a complex contagion dynamics.

This seems to really be the core of the paper, a comparison of simple and complex contagion. No one has done this before? The authors are FAR from the first to consider the two, so I find this statement unsupported. Seeing researchers debate between two different possibilities and then saying, "Sometimes it's both" isn't really impactful. Especially when it hinges entirely on two particular models--why not throw other complex contagion models like threshold models into the mix?

Readability.

I struggle to follow the manuscript. The authors should take a step back and focus on what message they are making. It is unfair to ask a reader to wade through this paper. It's also very hard to follow the figures. For example, Fig 2d, I gave up trying to keep track of all the dotted and dashed lines! Likewise, several figures in the SI don't seem to have captions that match their contents (S7 has many references to solid black lines that appear to not exist). This does not instill confidence.

Eq 5 seems to go out of its way to draw a connection with (negative) cross-entropy. Is this the best way to present the calculation? For one, it hides where the observed data enter into the likelihood.

Model selection.

I guess I am wrong in my comments about model selection, but I cannot tell for sure because the paper is so hard to read. It sure seems complicated. It seems unusual to me how many parameters are discussed and fitted to data when the authors then say there is only one parameter. The authors also appear to not provide any statistics on how many time series' are distinguished by LRTs vs. the original p-values.

Synthetic data.

The methods used here don't appear to work very well on ground truth data, so how trustworthy are

the results on real data?

Scaling collapse.

The authors discuss collapse of rescaled curves repeatedly. I see only qualitative support for this, from discussion of plots. In all cases I feel the evidence is weak, often with deviations between curves occurring over many orders of magnitude. Compared to the over-the-top quantitative analysis, this qualitative evidence undermines the core arguments.

Fig 3d the authors discuss breaking down the time series based on which model fits better and then see a scaling cross-over in one set only To my eye (again, eye: qualitative) I don't see much evidence for the difference. The BP curves are maybe a *touch* more rounded over, but I am reluctant to read too much into such a small difference.

Finally, another reviewer wrote:

> I particularly like how the results support a distinction between cascades on Twitter based on semantic, hinting that some themes cascade similarly to simple contagion, while other more controversial themes cascade with properties akin to complex contagion. I agree with the manuscript that such observation could have an important impact on our collective fight against the spread of misinformation on social media.

I just want to say I don't see any evidence supporting this whatsoever. There are some hand-wavy results around this one word cloud about music and tv, limited to one dataset only, but I do not see a pattern between the two models the authors investigate---There's lots of music and/or tv stuff in both columns. Perhaps there is something here about endogenous vs exogenous shocks (a la ref 33) but I can't really see it. Adding a table of the top 30 (twitter) hashtags doesn't really instill confidence. Something more systematic is needed. While the authors pitch these semantic notions as only part of the discussion, it appears in the abstract so it must be important. If the result is there, the authors should determine a research protocol to see if their hypothesis around semantics is true or not, and not merely leave it to discuss.

Reviewer #3 (Remarks to the Author):

I think the authors have addressed all my concerns. Browsing over the other reports I think they did a pretty good job on these as well. I recommend publication without further revisions in Nature Communications.

Best regards,
Baruch Barzel

First referee

I am satisfied by the authors' answers to my comments and to the modifications they made to the manuscript.

My recommendation is to accept the manuscript for publication once the second reviewer, whose background on the topic appears to be deeper than mine, is satisfied.

We thank the referee for the time devoted to our work. We are happy to know that the reviewer is recommending our work for publication.

Second referee

I have considered the revised manuscript. I appreciate the efforts to clarify their paper. However, I still struggle to see the importance of the study-peeling back all the statistical physics, the authors seem to be investigating simple versus complex contagion and reporting "sometimes it's both". Why is this impactful? Likewise, why is it impactful to find that social systems are distinct from earthquakes and rat brains? Despite the rebuttal, I remain skeptical of some arguments employed by the authors. Finally, I find the paper exceedingly hard to read, harder than before. This alone in my opinion makes it unsuitable for a general audience venue such as Nature Communications.

We appreciate the additional time that the referee dedicated to the manuscript. We did our best effort to address the concern raised below. On occasion, we found the comments too unspecific to prescribe specific actions. On other occasions, we found that we had already addressed the concern in the previous round of review. Hopefully, the present note will help resolve the communication issue we may have had with reviewer 2.

The reviewer is concerned about two main aspects of the paper. First, the lack of impact due to the finding that social systems display different properties of non-social systems. We believe that even a cursory reading of our paper can convince that this is not the case. The analysis of social systems is the only focus of the paper. Any reference to non-social systems has been removed from the main paper after its first revision.

The second point of concern of the reviewer regards our model comparison between simple vs. complex contagion models. It is not entirely clear to us whether the reviewer is mostly concerned with the relevance of the problem or the novelty/importance of our contribution.

As argued in the paper and above we maintain that discriminating between simple and complex contagion in regard of online information spreading is extremely important, both from a theoretical and application point of view. About our contribution: our paper studies statistical properties of large-scale data sets to demonstrate that information propagation in social media is compatible with a (1) universal and (2) critical process. Findings (1) and (2) should be quite apparent from the paper, starting already from its title. The paper concerns also (3) the comparison between simple vs. complex contagion models. However, we do not understand the technical concern of the reviewer. Our approach is in line with a well-established methodology in statistical model comparison, see for example Clauset *et al.* The fact that the application of this methodology reveals the coexistence of both mechanisms is a novel and important result. The referee seems to suggest that only finding that one of the two mechanisms strongly dominates the other would have been of interest. We regrettably feel in disagreement with the reviewer on this point.

In the discussion, the authors write,

“Also, in contrast with the vast majority of previous studies where purely diffusive models have been considered [37], we showed that information propagation in social media is often better described by a complex contagion dynamics.”

This seems to really be the core of the paper, a comparison of simple and complex contagion. No one has done this before? The authors are FAR from the first to consider the two, so I find this statement unsupported. Seeing researchers debate between two different possibilities and then saying, “Sometimes it’s both” isn’t really impactful. Especially when it hinges entirely on two particular models—why not throw other complex contagion models like threshold models into the mix?

We probably don’t need to restate our position about the significance of our work here, as we already stressed it above. We are not aware of any papers that study the problem at a scale and with methods similar to ours. We were eager to review the reference we asked for in our previous correspondence with the reviewer. Unfortunately, the nature of the reviewer’s critique remained too vague to be properly addressed.

We would like to stress that the Random Field Ising model, i.e., the model that we use here to represent the class of complex contagion, is indeed a threshold model. Adding other models to “the mix” would enrich our approach only if those belonged to different universality classes.

Readability.

I struggle to follow the manuscript. The authors should take a step back and focus on what message they are making. It is unfair to ask a reader to wade through this paper. It’s also very hard to follow the figures. For example, Fig 2d, I gave up trying to keep track of all the dotted and dashed lines! Likewise, several figures in the SI don’t seem to have captions that match their contents (S7 has many references to solid black lines that appear to not exist). This does not instill confidence.

We fixed the problem with the caption of Figure S7. Thanks for making this explicit pointer!

Figure 2d was changed following the advice of Referee 1 in the previous round of review. Reviewer 1 seems happy with all changes we performed, so we would prefer to keep the figure as it is. Note that Figure 2d summarizes information already presented in panels a, b and c of the same figure. Also if needed, numerical estimates of the critical exponents can be retrieved in Table S7.

Eq 5 seems to go out of its way to draw a connection with (negative) cross-entropy. Is this the best way to present the calculation? For one, it hides where the observed data enter into the likelihood.

As it is well known, the log-likelihood is equivalent to the negative cross-entropy, see for example https://en.wikipedia.org/wiki/Cross_entropy. We added a sentence in the Methods section of the paper to stress this fact.

Model selection. I guess I am wrong in my comments about model selection, but I cannot tell for sure because the paper is so hard to read. It sure seems complicated. It seems unusual to me how many parameters are discussed and fitted to data when the authors then say there is only one parameter. The authors also appear to not provide any statistics on how many time series' are distinguished by LRTs vs. the original p-values.

Both models rely on one parameter only. This fact has been already discussed in the previous correspondence, and addressed properly in our revised version of the manuscript.

Information about the distinction of time series based on LRT or p-value is explicitly reported in the bar chart of Figure 4a.

Synthetic data. The methods used here don't appear to work very well on ground truth data, so how trustworthy are the results on real data?

Our methods correctly pass the self-consistency test, see Figure S10.

Scaling collapse. The authors discuss collapse of rescaled curves repeatedly. I see only qualitative support for this, from discussion of plots. In all cases I feel the evidence is weak, often with deviations between curves occurring over many orders of magnitude. Compared to the over-the-top quantitative analysis, this qualitative evidence undermines the core arguments.

As already discussed in the previous round of review, universality is deduced from the fact that the critical exponent values measured over the various data sets are very similar, see Figure 2d and Table S7. We used the over-the-top quantitative method by Clauset *et al.* to determine the values of the critical exponents.

Fig 3d the authors discuss breaking down the time series based on which model fits better and then see a scaling cross-over in one set only. To my eye (again, eye: qualitative) I don't see much evidence for the difference. The BP curves are maybe a *touch* more rounded over, but I am reluctant to read too much into such a small difference.

In direct response to this comment, for which we are grateful, we decided to quantify the difference between the two scaling regimes. We considered the same data as in Figure 3d of the main paper, and divided the sample of avalanche size in two subsamples, one with the avalanches with size $2 \leq S \leq 700$ and the other with size $700 < S$. The value $S = 700$ approximately marks the crossover point, see Figure R1. We then computed the maximum likelihood estimator of the exponent τ for the two subsamples. Results are shown in Table R1. The statistical analysis confirms the existence of two different scaling regimes. We note that the two statistical regimes are consistent with the predictions of the branching process (small avalanches subsample) and of the Random Field Ising Model (large avalanches subsample). Further, different systems display similar values of the estimator, confirming that the behaviour of these systems is universal.

System	$\hat{\tau}$ for $2 \leq S \leq 700$	$\hat{\tau}$ for $700 < S$
TWT	1.7092 ± 0.0005	2.095 ± 0.005
TLG	1.437 ± 0.002	2.34 ± 0.04
PARL	1.482 ± 0.003	2.18 ± 0.05
WEI	1.495 ± 0.004	2.44 ± 0.08
STCK	1.384 ± 0.002	2.15 ± 0.03
DEL	1.404 ± 0.005	1.98 ± 0.06

Table R1: We display the maximum likelihood estimator of the exponent τ for the two subsamples of avalanches that are obtained from the data in Figure 3d of the main paper, specifically the data that are classified as nearly critical (relative distance from the critical point $|\hat{n} - n_c|/n < 0.05$) branching processes. Values of $\hat{\tau}$ corresponding to the regimes $2 \leq S \leq 700$ and $700 < S$ are reported in the table.

Figure R1: We display the same distributions as in Figure 3d of the main paper for avalanches classified as nearly critical (relative distance from the critical point $|\hat{n} - n_c|/n < 0.05$) branching processes. The vertical grey line marks $S = 700$, roughly coinciding with the crossover point. The dotted lines correspond to best fit of the exponent $\hat{\tau}$ to the RFIM (red) and to the BP (teal), respectively.

Finally, another reviewer wrote:

“I particularly like how the results support a distinction between cascades on Twitter based on semantic, hinting that some themes cascade similarly to simple contagion, while other more controversial themes cascade with properties akin to complex contagion. I agree with the manuscript that such observation could have an important impact on our collective fight against the spread of misinformation on social media.”

I just want to say I don't see any evidence supporting this whatsoever. There are some hand-wavy results around this one word cloud about music and tv, limited to one dataset only, but I do not see a pattern between the two models the authors investigate—There's lots of music and/or tv stuff in both columns. Perhaps there is something here about endogenous vs exogenous shocks (a la ref 33) but I can't really see it. Adding a table of the top 30 (twitter) hashtags doesn't really instill confidence. Something more systematic is needed. While the authors pitch these semantic notions as only part of the discussion, it appears in the abstract so it must be important. If the result is there, the authors should determine a research protocol to see if their hypothesis around semantics is true or not, and not merely leave it to discuss.

We addressed this issue in response to a comment by reviewer 3 which is apparently satisfied by our response. Our speculation is supported by previous literature, see for example Romero *et al.* (i.e., Ref.[45] in our paper) and somewhat resonate with some of the five points identified by Macy and Centola as basic mechanisms for complex contagions [Centola and Macy, Complex Contagions and the Weakness of Long Ties, *American Journal of Sociology*, Vol. 113, No. 3 (November 2007), pp. 702-734]. We acknowledge that the last sentence of the abstract was probably too strong in light of a point raised only in the discussion to motivate further investigation and hence we removed it.

In general, we agree with the reviewer that an analysis aimed at relating endogenous vs. exogenous shocks to the classification of time series in simple vs. complex contagion classes could be a good topic of research for follow-up papers.

Third referee

I think the authors have addressed all my concerns. Browsing over the other reports I think they did a pretty good job on these as well. I recommend publication without further revisions in Nature Communications.

We are grateful to the reviewer for the time dedicated to our manuscript and for recommending our work for publication in Nature Communications.

List of changes

1. We fixed the issue with the caption of Figure S7 and Figure S8.
2. We removed the last sentence from the abstract.
3. We added a sentence to the method section, just below Eq. (5): “The maximization of the log-likelihood ... to the distribution P ”.
4. We updated the affiliation of one of our authors (Castellano Claudio).
5. We reformatted the manuscript following as closely as possible the editorial instructions.